# MECHANISTIC INDEPENDENCE: A PRINCIPLE FOR IDENTIFIABLE DISENTANGLED REPRESENTATIONS

**Stefan Matthes**
Technical University of Munich
`stefan.matthes@tum.de`

**Zhiwei Han**
Technical University of Munich
`zhiwei.han@tum.de`

**Hao Shen**
Technical University of Munich
`hao.shen@tum.de`

## ABSTRACT

*Disentangled representations* seek to recover latent factors of variation underlying observed data, yet their *identifiability* is still not fully understood. We introduce a unified framework in which disentanglement is achieved through *mechanistic independence*, which characterizes latent factors by how they act on observed variables rather than by their latent distribution. This perspective is invariant to changes of the latent density, even when such changes induce statistical dependencies among factors. Within this framework, we propose several related independence criteria – ranging from support-based and sparsity-based to higher-order conditions – and show that each yields identifiability of latent subspaces, even under nonlinear, non-invertible mixing. We further establish a hierarchy among these criteria and provide a graph-theoretic characterization of latent subspaces as connected components. Together, these results clarify the conditions under which disentangled representations can be identified without relying on statistical assumptions.

## 1 INTRODUCTION

Disentangled representations capture the underlying explanatory factors that generate observed data. They are widely believed to promote compositionality, enable controllable generation, and facilitate transfer (Bengio et al., 2013; Higgins et al., 2017; Schölkopf et al., 2021; Locatello et al., 2019; Greff et al., 2020; Goyal & Bengio, 2022)). From a scientific perspective, disentanglement aligns with the goal of discovering the causal or mechanistic structure of data-generating processes (Schölkopf et al., 2021). The question of whether such representations can be consistently recovered is addressed by identifiability. If a model class lacks identifiability, different training runs may encode incompatible factors, thereby undermining interpretability and transfer.

A classical route to identifiability is to posit *statistical independence* of the latent factors, as in independent component analysis (ICA) (Comon, 1994; Hyvärinen & Oja, 2000) and independent subspace analysis (ISA) (Cardoso, 1998; Hyvärinen & Hoyer, 2000). Early work focused on linear mixing, where identifiability can be obtained under mild conditions. For general *nonlinear* mixing, however, identifiability is impossible without further assumptions (Hyvärinen & Pajunen, 1999; Locatello et al., 2019), motivating a large body of work that augments statistical assumptions with temporal cues (Hyvärinen & Morioka, 2016; 2017; Klindt et al., 2020), auxiliary variables (Hyvärinen & Morioka, 2017; Hyvärinen et al., 2019; Khemakhem et al., 2020a), multiple views (Khemakhem et al., 2020b; Gresele et al., 2020; Von Kügelgen et al., 2021; Zimmermann et al., 2021; Matthes et al., 2023), or interventions (Locatello et al., 2020; Lachapelle et al., 2022; Ahuja et al., 2022; Brehmer et al., 2022; Ahuja et al., 2023; Jiang & Aragam, 2023; Yao et al., 2023; Zhang et al., 2024; Ng et al., 2025).

A complementary strategy constrains the *mechanism* that maps latents to observations, such as post-nonlinear mixtures (Taleb & Jutten, 1999; Jutten & Karhunen, 2004; Zhang & Hyvärinen, 2012; Lyu & Fu, 2021), near-linear mixtures (Zhang et al., 2008), local isometries (Horan et al., 2021),

piecewise-affine mixtures (Kivva et al., 2022), sparsity (Moran et al., 2021; Zheng et al., 2022; Zheng & Zhang, 2023), additive structures (Lachapelle et al., 2023), conformal maps and orthogonal coordinate transformations (Gresele et al., 2021; Buchholz et al., 2022; Ghosh et al., 2023), and Jacobian-based constraints (Brady et al., 2023; 2024; Nguyen & Fu, 2025). Independent Mechanism Analysis (IMA) (Gresele et al., 2021) proposes to address nonlinear ICA by restricting the mixing function so that its Jacobian has orthogonal columns. This couples statistical independence of the latents with a mechanistic constraint on the generator. In contrast, we pursue *mechanistic independence* as a stand-alone organizing principle: factors are defined by *how they act* on observations (through the generator), not by how they are distributed. This shift yields identifiability statements that are invariant to reweightings of the latent density and allows the true factors to be misaligned with any statistically independent subspaces.

This work presents a family of mechanistic independence criteria – spanning support-based separation, sparsity gaps in first-order action, and higher-order (cross-derivative) constraints. Similar to ISA that shows identifiability with respect to a minimal decomposition into independent subspaces, each criterion comes with a corresponding notion of *irreducibility* that rules out spurious internal splits of a factor and yields an identifiability theorem. Our framework covers multi-dimensional factors and partial disentanglement.

Our framework generalizes and unifies recent identifiability results based on mechanistic constraints: object-centric disentanglement via disjoint supports (Brady et al., 2023), interaction asymmetry (Brady et al., 2024), and additive decoders (Lachapelle et al., 2023), and it partially subsumes sparsity-based nonlinear ICA results (Zheng et al., 2022; Zheng & Zhang, 2023) (the parts that do not require statistical independence). Moreover, defining independent mechanisms by Jacobian-orthogonality as in IMA (Gresele et al., 2021) appears in our taxonomy as one instance within a broader class of mechanistic constraints. Unlike approaches that rely primarily on distributional assumptions (e.g., temporal structure or auxiliary variables), our results hinge on properties of the generator and therefore remain valid under broad latent densities. The main contributions of this work are as follows.

- We define a notion of local disentanglement and prove that under mild topological assumptions (such as path-connectedness of the source space) local disentanglement extends to global disentanglement even for generators that are not fully invertible.
- We introduce a family of mechanistic independence criteria for subspaces and prove for each identifiability (up to block-wise invertible transforms and permutations).
- We discuss how the independence criteria are related and show that the independent and irreducible factors coincide with connected components of graphs derived from mechanistic assumptions of the generator.

**Notation** We write $[n] \coloneqq \{1, \dots, n\}$ for $n \in \mathbb{N}$. Scalars are denoted by lowercase letters, vectors by bold lowercase, and matrices by bold uppercase (e.g., $a \in \mathbb{R}$, $\boldsymbol{a} \in \mathbb{R}^n$, $\boldsymbol{A} \in \mathbb{R}^{n \times n}$). Scalar-valued functions are written $f, f_i$, while general maps are written $\boldsymbol{f}, \boldsymbol{f}_i$. For $\boldsymbol{p} \in \mathcal{S}_1 \times \cdots \times \mathcal{S}_n$, we set $D_i \boldsymbol{f}_{\boldsymbol{p}} \coloneqq D\boldsymbol{f}_{\boldsymbol{p}} \circ \iota_i$ for the differential in the $i$-th argument ($\iota_i$ the canonical inclusion), and more generally $D_{i,j}^n \boldsymbol{f}_{\boldsymbol{p}} \coloneqq D^n \boldsymbol{f}_{\boldsymbol{p}} \circ (\iota_i, \iota_j, \mathrm{id}, \dots, \mathrm{id})$.

## 2 Disentanglement and Identifiability

We now formalize the data-generating assumptions and the notion of disentanglement used throughout the paper, before turning to identifiability. Our goal is to explain when a decoder (or encoder) *recovers*, up to natural ambiguities, the underlying factors of variation that compose the observations.

### 2.1 Data Generating Process

We model latent factors of variation as subspaces of a product manifold, reflecting the often compositional nature of observed data. Let the set of generative (latent) configurations be an open[1] subset

---

[1] The condition that $\mathcal{S}$ is open implies that each factor can vary independently and without restriction at any point within the space and is a common assumption.

$\mathcal{S} \subseteq \mathcal{S}_1 \times \cdots \times \mathcal{S}_K$, where each factor space $\mathcal{S}_i$ has positive dimension. We assume the latent distribution $\mathbb{P}_{\boldsymbol{s}}$ is strictly positive on $\mathcal{S}$.

In line with the manifold hypothesis in representation learning (though assuming that observations lie on rather than merely near a manifold), we posit that observations are produced via a *generator* (also called a *ground-truth decoder* or *mixing function*)

$$\boldsymbol{g} \colon \mathcal{S} \to \mathcal{X} \subseteq \mathbb{R}^{d_x}.$$

We denote the observation manifold by $\mathcal{X} \coloneqq \boldsymbol{g}(\mathcal{S})$, where typically $d_s \coloneqq \dim(\mathcal{S})$ is much smaller than $d_x$.

Notably, instead of characterizing the underlying factors through (conditional) statistical independence or latent-space group actions (Higgins et al., 2018), we characterize them by their action on the observation manifold via $\boldsymbol{g}$. While these notions may align, they do not necessarily have to. Several possibilities for making this precise are discussed in Section 3.

## 2.2 DISENTANGLED REPRESENTATIONS

To discuss how a learned representation may or may not reflect the underlying generative factors, we consider a target representation space $\mathcal{Z} \subseteq \prod_{j=1}^{L} \mathcal{Z}_j$. In a disentangled representation, each component $\mathcal{Z}_j$ is intended to capture a single generative factor, or at most a restricted subset of factors. We formalize this with the notion of a *decomposable map*.

**Definition 1** (Decomposable map). *Let $\mathcal{S} \subseteq \prod_{i=1}^{K} \mathcal{S}_i$ and $\mathcal{Z} \subseteq \prod_{j=1}^{L} \mathcal{Z}_j$. A map $\boldsymbol{h} \colon \mathcal{S} \to \mathcal{Z}$ is partially decomposable if there are a surjection $\sigma \colon [K] \to [L]$ and maps $\boldsymbol{h}^{(j)} \colon \prod_{i \in \sigma^{-1}(j)} \mathcal{S}_i \to \mathcal{Z}_j$ such that, for all $\boldsymbol{s} \in \mathcal{S}$,*

$$\boldsymbol{h}(\boldsymbol{s}) = \left( \boldsymbol{h}^{(j)}((\boldsymbol{s}_i)_{i \in \sigma^{-1}(j)}) \right)_{j=1}^{L}. \tag{1}$$

*If $K = L$, it is called fully decomposable, or simply decomposable.*

Thus, each target factor $\boldsymbol{z}_j \in \mathcal{Z}_j$ depends only on the source factors $\{\boldsymbol{s}_i : \sigma(i) = j\}$.

**Definition 2** (Disentanglement). *A decoder $\hat{\boldsymbol{g}} \colon \mathcal{Z} \to \mathcal{X}$ is disentangled w.r.t. a generator $\boldsymbol{g} \colon \mathcal{S} \to \mathcal{X}$ if $\boldsymbol{g} = \hat{\boldsymbol{g}} \circ \boldsymbol{h}$ for some decomposable map $\boldsymbol{h} \colon \mathcal{S} \to \mathcal{Z}$. If $\boldsymbol{h}$ is partially decomposable, this is called partial disentanglement.*

Disentanglement asserts that varying a single factor of the learned representation changes the decoded observation exactly as varying the corresponding source factors would. It can also be defined in terms of an encoder $\hat{\boldsymbol{f}} \colon \mathcal{X} \to \mathcal{Z}$ (e.g., $\hat{\boldsymbol{f}} \circ \boldsymbol{g} = \boldsymbol{h}$). However, when $\boldsymbol{g}$ is not invertible, $\hat{\boldsymbol{f}}$ may not exist or may lack desirable properties such as continuity[2]. Notably, an oracle generator would be trivially disentangled w.r.t. itself, even if not invertible. Under additional regularity assumptions, disentanglement forms an equivalence relation (see Propositions 1 and 2), meaning that $\boldsymbol{g}$ and $\hat{\boldsymbol{g}}$ represent equivalent generative models.

More generally, $\hat{\boldsymbol{g}}$ is *locally disentangled* if, for every $\boldsymbol{s} \in \mathcal{S}$, there exists a neighborhood of $\boldsymbol{s}$ where the restriction of $\boldsymbol{g}$ admits such a disentangled representation (see Defn. 11). At first glance, local disentanglement may appear less significant than the global property. However, under mild topological constraints the two notions coincide, even when $\boldsymbol{g}$ is not fully invertible (see next section).

## 2.3 IDENTIFIABILITY

Identifiability asks whether a (locally) disentangled description is essentially unique given only observations in $\mathcal{X}$. It characterizes when a learned representation must be disentangled. The following global result shows that, under mild topological assumptions, local disentanglement implies global disentanglement. The key condition is connectedness of slices in the source space. A $k$-slice is the subspace obtained by holding all but $k$ factors constant (see Defn. 12). Note that path-connectedness of a space and of its slices are related but independent notions (see Remark 1).

---

[2]A practical example where continuity breaks is the *responsibility problem* which arises when learning representations of unordered data, such as sets or objects within an image (Zhang et al., 2019; Hayes et al., 2023; Mansouri et al., 2023). The permutation invariance makes the generator non-invertible.

**Theorem 1** (Global identifiability). *Let $\mathcal{S}$ be an open subspace of the product manifold $\prod_{i=1}^{K} \mathcal{S}_i$, where each factor $\mathcal{S}_i$ has positive dimension. Assume $\hat{\boldsymbol{g}} : \mathcal{Z} \to \mathcal{X}$ is locally disentangled w.r.t. $\boldsymbol{g} : \mathcal{S} \to \mathcal{X}$. Then $\hat{\boldsymbol{g}}$ is globally disentanglement if:*

*(1) $\mathcal{S}$ is simply connected[3].*

*(2) Every (K-1)-slice of $\mathcal{S}$ is path-connected.*

*(3) $\boldsymbol{g} : \mathcal{S} \to \mathcal{X}$ is continuous and locally injective.*

*(4) $\hat{\boldsymbol{g}} : \mathcal{Z} \to \mathcal{X}$ is a covering map[4].*

A proof is given in Appendix A.1. Informally, local disentanglement propagates along paths: since each factor can vary independently, and local injectivity prevents branching, local decompositions extend globally. Conditions (1) and (4) can be further relaxed (see Remark 2).

In many practical cases (e.g., convex open sets in $\mathbb{R}^n$), the topological conditions on $\mathcal{S}$ hold automatically. Thus, the main challenge is usually to establish local disentanglement, and the remainder of the paper therefore focuses on local identifiability.

## 3 IDENTIFIABILITY VIA INDEPENDENT MECHANISMS

We now establish a general framework that certifies local disentanglement by analyzing how latent factors *act* on the observation manifold through the generator $\boldsymbol{g}$. The key difference from classical approaches is that independence is formulated at the level of the *generative mechanism* rather than the latent probability law. As a result, it accommodates almost arbitrary distributions, including those with statistical dependencies between and within subspaces. Importantly, there is no universal notion of mechanistic independence comparable to statistical independence. Instead, we present a family of independence criteria – disjointedness (Type D), mutual non-inclusion (Type M), sparsity gap (Type S), and higher-order separability (Type H$_n$) – each of which leads to disentangled representations when mirrored in the learned representation.

### 3.1 LOCAL IDENTIFIABILITY OF TYPE D

We begin by slightly extending the result of Brady et al. (2023) and rephrasing it within our framework.

**Definition 3** (Mechanistic independence of Type D). *We say that $\mathcal{S}_i$ and $\mathcal{S}_j$ (equivalently, $\boldsymbol{s}_i$ and $\boldsymbol{s}_j$) are mechanistically independent of Type D if, for all $\boldsymbol{s} \in \mathcal{S}$, $\boldsymbol{u} \in T_{\boldsymbol{s}_i}\mathcal{S}_i$, and $\boldsymbol{v} \in T_{\boldsymbol{s}_j}\mathcal{S}_j$,*

$$D_i \boldsymbol{g}_{\boldsymbol{s}}(\boldsymbol{u}) \bullet D_j \boldsymbol{g}_{\boldsymbol{s}}(\boldsymbol{v}) = \boldsymbol{0}, \tag{2}$$

*where $T_{\boldsymbol{s}_i}\mathcal{S}_i$ denotes the tangent space of $\mathcal{S}_i$ at $\boldsymbol{s}_i$ and $\bullet$ denotes the element-wise (Hadamard) product in $\mathbb{R}^{d_x}$.[5]*

We call this Type D independence since Hadamard orthogonality expresses that different factors act on a *disjoint* set of observation coordinates. For example, in images, each factor controls a non-overlapping set of pixels. Independence among the $\mathcal{Z}_j$ is defined analogously via $\hat{\boldsymbol{g}}$.

To ensure disentanglement, independence alone is insufficient: if a source factor $\mathcal{S}_i$ can be decomposed into smaller, mutually independent components, a learned representation may split and recombine them arbitrarily. This motivates the notion of reducibility.

**Definition 4** (Reducibility of Type D). *We say that $\mathcal{S}_i$ is reducible of Type D if there exists $\boldsymbol{s} \in \mathcal{S}$ such that $T_{\boldsymbol{s}_i}\mathcal{S}_i$ admits a nontrivial[6] direct-sum decomposition $T_{\boldsymbol{s}_i}\mathcal{S}_i = U \oplus V$ with the property that, for all $\boldsymbol{u} \in U$ and $\boldsymbol{v} \in V$,*

$$D_i \boldsymbol{g}_{\boldsymbol{s}}(\boldsymbol{u}) \bullet D_i \boldsymbol{g}_{\boldsymbol{s}}(\boldsymbol{v}) = \boldsymbol{0}.$$

*If no such decomposition exists, we call $\mathcal{S}_i$ irreducible of Type D.*

---

[3] A simply connected set is path-connected and such that every loop can be continuously shrunk to a point.

[4] A covering map is a continuous map such that every point of the codomain has a neighborhood whose preimage is a disjoint union of open sets, each mapped homeomorphically onto that neighborhood.

[5] Throughout this work, we identify $T_{\boldsymbol{g}(\boldsymbol{s})}\mathcal{X}$ with its natural inclusion in $\mathbb{R}^{d_x}$.

[6] "Nontrivial" means $\dim(U), \dim(V) > 0$.

This coincides with reducibility as defined in (Brady et al., 2023) (see Proposition 5), but makes the connection to Type D independence explicit. If $\mathcal{S}_i$ is reducible we could split it at a point into smaller independent subspaces, and if a factor is one-dimensional it should always be irreducible.

**Theorem 2** (Local identifiability of Type D). *Let $\boldsymbol{g}\colon \mathcal{S} \to \mathcal{X}$ and $\hat{\boldsymbol{g}}\colon \mathcal{Z} \to \mathcal{X}$ be local diffeomorphisms[7] with $\boldsymbol{g}(\mathcal{S}) \subseteq \hat{\boldsymbol{g}}(\mathcal{Z})$. Then $\hat{\boldsymbol{g}}$ is locally disentangled w.r.t. $\boldsymbol{g}$ if:*

*(1) $\mathcal{S} \subseteq \prod_{i=1}^{K} \mathcal{S}_i$ is open, and all factors are Type D independent and irreducible.*

*(2) $\mathcal{Z} \subseteq \prod_{i=1}^{L} \mathcal{Z}_i$ is open with $L = K$, and the factors are independent of Type D.*

*If $L < K$, then $\hat{\boldsymbol{g}}$ is locally partially disentangled.*

Intuitively, if each source factor influences a disjoint set of observation coordinates, and no finer decomposition is possible, then any learned representation that also acts on disjoint coordinates recovers the true source factors (up to block-wise invertible transformations and permutations).

This result generalizes Theorem 1 of (Brady et al., 2023) to partial disentanglement and non-invertible generators (when taking Theorem 1 into account). A proof is given in Appendix A.3. Interestingly, all local identifiability proofs in this paper follow a common template: starting from the local reconstruction identity $\hat{\boldsymbol{g}} = \boldsymbol{g} \circ \boldsymbol{v}$ (where $\boldsymbol{v} \coloneqq \boldsymbol{g}^{-1} \circ \hat{\boldsymbol{g}}$ exists locally since both maps are local diffeomorphisms), one applies the independence conditions to constrain interactions between source and target factors. If a source factor interacted with multiple target factors, their independence would force a decomposition of the source factor, contradicting irreducibility. Occasionally, additional assumptions are needed to further restrict the function class.

Since Type D independence requires that no observation coordinate is affected by two factors, a natural question is whether this can be relaxed to allow limited overlap while still achieving identifiability. We next express this via supports (the index set of nonzero elements, denoted with $\mathrm{supp}(\cdot)$) of Jacobians.

Select a product basis $(\boldsymbol{u}_1, \ldots, \boldsymbol{u}_{d_s})$ for $T_{\boldsymbol{s}}\mathcal{S}$; define $\Omega_i(\boldsymbol{s}) \coloneqq \mathrm{supp}(D\boldsymbol{g}_{\boldsymbol{s}}(\boldsymbol{u}_i))$ for the $i$-th basis vector; and let $\mathcal{C}_j$ be the index set of basis vectors of $T_{\boldsymbol{s}_j}\mathcal{S}_j$. Then Type D independence can be reformulated as

$$\forall i \neq j, \forall a \in \mathcal{C}_i, \forall b \in \mathcal{C}_j : \quad \Omega_a(\boldsymbol{s}) \cap \Omega_b(\boldsymbol{s}) = \varnothing. \tag{3}$$

As long as the basis respects the product structure, the particular choice does not matter. In the next two sections, we show how this condition can be relaxed, either via *mutual non-inclusion* or through a *sparsity gap*.

## 3.2 LOCAL IDENTIFIABILITY OF TYPE M

Define the *mutual non-inclusion* relation between sets $\mathcal{A}, \mathcal{B} \subseteq [d_x]$ as $\mathcal{A} \pitchfork \mathcal{B} \coloneqq \mathcal{A} \nsubseteq \mathcal{B} \wedge \mathcal{A} \nsupseteq \mathcal{B}$, that is, the sets may intersect, but neither is contained in the other.

**Definition 5** (Mechanistic independence of Type M). *We say that $\mathcal{S}_i$ and $\mathcal{S}_j$ are mechanistically independent of Type M if, for every $\boldsymbol{s} \in \mathcal{S}$,*

$$\forall i \neq j, \forall a \in \mathcal{C}_i, \forall b \in \mathcal{C}_j : \quad \Omega_a(\boldsymbol{s}) \pitchfork \Omega_b(\boldsymbol{s}). \tag{4}$$

Type M independence allows observation coordinates to be influenced jointly by multiple factors as long as neither support fully contains the other. In image data, for example, different factors may affect intersecting sets of pixels, allowing partial occlusion, shadows and reflections. Unlike Type D independence, this notion depends on the choice of basis for $T_{\boldsymbol{s}}\mathcal{S}$. To make it meaningful, we restrict to $\mathcal{S} \subseteq \mathbb{R}^{d_s}$ (only for Type M), where $T_{\boldsymbol{s}}\mathbb{R}^{d_s}$ carries a canonical basis that aligns with the product structure. Reducibility is then expressed directly in these fixed coordinates.

**Definition 6** (Reducibility of Type M). *The component $\mathcal{S}_i$ is reducible of Type M if there exist $\boldsymbol{s} \in \mathcal{S}$ and a partition $\mathcal{C}_i = \mathcal{A} \cup \mathcal{B}$ such that*

$$\forall a \in \mathcal{A}, \forall b \in \mathcal{B} : \quad \Omega_a(\boldsymbol{s}) \pitchfork \Omega_b(\boldsymbol{s}).$$

---

[7]A diffeomorphism is a smooth bijection between manifolds with a smooth inverse. A local diffeomorphism is a map that restricts to a diffeomorphism on some neighborhood of each point.

**Theorem 3** (Local identifiability of Type M). *Let $\boldsymbol{g} \colon \mathcal{S} \to \mathcal{X}$ and $\hat{\boldsymbol{g}} \colon \mathcal{Z} \to \mathcal{X}$ be local diffeomorphisms with $\boldsymbol{g}(\mathcal{S}) \subseteq \hat{\boldsymbol{g}}(\mathcal{Z})$. Then $\hat{\boldsymbol{g}}$ is locally disentangled w.r.t. $\boldsymbol{g}$ if:*

*(1) $\mathcal{S} \subseteq \mathbb{R}^{d_s}$ is open, and the factors are Type M independent and irreducible.*

*(2) $\mathcal{Z} \subseteq \mathbb{R}^{d_s}$ is open, and the factors are independent of Type M.*

*(3) For all $\boldsymbol{s} \in \mathcal{S}$ and $\boldsymbol{z} \in \mathcal{Z}$ with $\boldsymbol{g}(\boldsymbol{s}) = \hat{\boldsymbol{g}}(\boldsymbol{z})$,*

$$\|\boldsymbol{J}_{\hat{\boldsymbol{g}}}(\boldsymbol{z})\|_0 \leq \|\boldsymbol{J}_{\boldsymbol{g}}(\boldsymbol{s})\|_0. \tag{5}$$

*(4) For all such pairs,*

$$\widehat{\Omega}_k(\boldsymbol{z}) = \bigcup_{i \in \mathrm{supp}(\boldsymbol{B}_{:,k})} \Omega_i(\boldsymbol{s}), \tag{6}$$

*where $\boldsymbol{B} := \boldsymbol{J}_{\boldsymbol{g}^{-1} \circ \hat{\boldsymbol{g}}}(\boldsymbol{z})$ and $\widehat{\Omega}_k$ mirrors $\Omega_i$ for $\hat{\boldsymbol{g}}$.*

This theorem generalizes Theorem 3.1 of (Zheng & Zhang, 2023) (itself an extension of (Zheng et al., 2022)) to multidimensional factors (see Proposition 4 for a detailed comparison). Statistical independence of the sources is not required. Assumptions (1)–(2) mirror those in Theorem 2; condition (3) motivates a sparsity regularizer; and condition (4) rules out pathological cases and is implied by condition *(i)* in (Zheng & Zhang, 2023). It usually holds when $\boldsymbol{g}$ is sufficiently nonlinear, though a failure mode is illustrated in Example 1, case $\boldsymbol{B}$, where the Jacobian is constant on $\mathcal{S}$.

### 3.3 LOCAL IDENTIFIABILITY OF TYPE S

We now return to the setting where $\mathcal{S}$ is a smooth manifold and replace the mutual non-inclusion assumption with a *sparsity gap* criterion. Among all coordinate systems, the basis aligned with the true factor decomposition yields the sparsest first-order action of the generator.

For $\boldsymbol{s} \in \mathcal{S}$, let $\rho_{\mathfrak{B}}^+(\boldsymbol{s})$ be the minimal $\ell_0$-norm of the matrix representing $D\boldsymbol{g}_{\boldsymbol{s}} \colon T_{\boldsymbol{s}}\mathcal{S} \to T_{\boldsymbol{g}(\boldsymbol{s})}\mathcal{X}$ when the domain basis is aligned with the decomposition

$$\mathfrak{B} := \bigoplus_{i \in [K]} T_{\boldsymbol{s}_i}\mathcal{S}_i.$$

Conversely, let $\rho_{\mathfrak{B}}^-(\boldsymbol{s})$ be the infimum of the $\ell_0$-norm over all bases of $T_{\boldsymbol{s}}\mathcal{S}$ that *do not* respect $\mathfrak{B}$.

**Definition 7** (Mechanistic independence of Type S). *The subspaces $\{\mathcal{S}_i\}_{i=1}^K$ are mechanistically independent of Type S if, for every $\boldsymbol{s} \in \mathcal{S}$,*

$$\rho_{\mathfrak{B}}^+(\boldsymbol{s}) \; < \; \rho_{\mathfrak{B}}^-(\boldsymbol{s}). \tag{7}$$

Viewing the Jacobian as a dictionary that maps infinitesimal latent directions to observation directions, Type S independence states that the sparsest such dictionary (in the $\ell_0$ sense) is attained precisely when the basis aligns with the true factorization. Any misalignment necessarily incurs a strict sparsity gap.

If the supports of different components are disjoint, any mixing of partial derivatives can only enlarge the support, since no cancellations are possible. In this case, Equation 7 holds trivially. Thus Type D independence is a special case of Type S independence. The sparsity gap, however, is considerably stronger: it remains valid even when the supports substantially overlap. For instance, suppose we have one-dimensional sources where each support $\Omega_i(\boldsymbol{s})$ overlaps with the others by less than half of its elements. Even if a misaligned basis were tuned so that every shared element canceled perfectly (if at all possible), the total number of nonzeros would still increase. Thus, the sparsity gap persists under this optimal misaligned (but still suboptimal) basis transformation. In higher-dimensional subspaces, the situation becomes more intricate, since inter-cancellations within block columns are possible. In a sense, the sparsity gap captures all such potential cancellations and characterizes the theoretical limiting case. As before, irreducibility rules out internal decompositions (see Defn. 18). Example 1 discusses Type M/S independence and reducibility in detail.

**Theorem 4** (Local identifiability of Type S). *Let $\boldsymbol{g} \colon \mathcal{S} \to \mathcal{X}$ and $\hat{\boldsymbol{g}} \colon \mathcal{Z} \to \mathcal{X}$ be local diffeomorphisms with $\boldsymbol{g}(\mathcal{S}) \subseteq \hat{\boldsymbol{g}}(\mathcal{Z})$. Then $\hat{\boldsymbol{g}}$ is locally disentangled w.r.t. $\boldsymbol{g}$ if:*

*(1)* $\mathcal{S} \subseteq \prod_{i=1}^{K} \mathcal{S}_i$ *is open, and the factors* $\mathcal{S}_i$ *are Type S independent and irreducible.*

*(2)* $\mathcal{Z} \subseteq \prod_{j=1}^{L} \mathcal{Z}_j$ *is open with* $L = K$*, and the factors* $\mathcal{Z}_j$ *are independent of Type S.*

*If* $L < K$*, then* $\hat{\boldsymbol{g}}$ *is locally partially disentangled.*

Intuitively, identifiability follows by exploiting the strict sparsity gap in equation 7. While fairly general, Equation 7 is intractable to optimize in practice. In Section 5 we investigate whether *compositional contrast* (Brady et al., 2023) can serve as a suitable surrogate loss.

## 3.4 LOCAL IDENTIFIABILITY OF TYPE H

Lastly, we simplify and generalize the asymmetric interaction principle of (Brady et al., 2024), subsuming as a special case the additive setting of (Lachapelle et al., 2023).

**Definition 8** (Mechanistic independence of Type $H_n$)**.** *Let* $\mathcal{S} \subseteq \prod_{i=1}^{K} \mathcal{S}_i$ *be a smooth manifold, and let* $\boldsymbol{g} \colon \mathcal{S} \to \mathcal{X}$ *be of class* $C^n$ *with* $n \geq 2$*. We say that* $\mathcal{S}_i$ *and* $\mathcal{S}_j$ *are* mechanistically independent *of Type* $H_n$ *if, for all* $\boldsymbol{s} \in \mathcal{S}$,

$$D_{i,j}^n \boldsymbol{g_s} = \boldsymbol{0}. \tag{8}$$

For $n = 2$, this requires that all cross-Hessian blocks vanish, implying additivity as in (Lachapelle et al., 2023). Irreducibility is defined analogously (see Defn. 20).

To derive disentanglement, we additionally constrain the function class via *separability*.

**Definition 9** (Separability of $n$-th order)**.** *We say that* $\boldsymbol{g} \colon \mathcal{S} \to \mathcal{X}$ *is separable of order* $n \geq 2$ *if there exists* $\boldsymbol{s} \in \mathcal{S}$ *such that, for all* $i \in [K]$*, the image of* $D_{i,i}^n \boldsymbol{g_s}$ *intersects trivially with*

$$\mathrm{span}\Big\{ D_{j,j}^n \boldsymbol{g_s}, \ j \neq i; \ D^k \boldsymbol{g_s}, \ 1 \leq k \leq n-1 \Big\}.$$

Separability is closely related to *sufficient independence* in (Brady et al., 2024) and *sufficient nonlinearity* in (Lachapelle et al., 2023), but is slightly weaker: it allows arbitrary interactions among lower-order derivatives and within each block $D_{i,i}^n \boldsymbol{g_s}$.

**Theorem 5** (Local identifiability of Type $H_n$)**.** *Let* $\boldsymbol{g} \colon \mathcal{S} \to \mathcal{X}$ *and* $\hat{\boldsymbol{g}} \colon \mathcal{Z} \to \mathcal{X}$ *be local* $C^n$*-diffeomorphisms with* $n \geq 2$ *satisfying* $\boldsymbol{g}(\mathcal{S}) \subseteq \hat{\boldsymbol{g}}(\mathcal{Z})$*. Then* $\hat{\boldsymbol{g}}$ *is locally disentangled w.r.t.* $\boldsymbol{g}$ *if:*

*(1)* $\mathcal{S} \subseteq \prod_{i=1}^{K} \mathcal{S}_i$ *is open, and the factors are Type* $H_n$ *independent and irreducible.*

*(2)* $\mathcal{Z} \subseteq \prod_{j=1}^{L} \mathcal{Z}_j$ *is open with* $L = K$*, and the factors are independent of Type* $H_n$*.*

*(3)* $\boldsymbol{g}$ *is separable of order* $n$*.*

*If* $L < K$*, then* $\hat{\boldsymbol{g}}$ *is locally partially disentangled.*

Compared to (Brady et al., 2024), our formulation highlights that source factors should be taken as irreducible, which we argue is a necessary and natural requirement. This perspective eliminates any dependence on $(n+1)$-th derivatives (which may not exist) and avoids the use of *equivalent generators*. As with our other results, the conclusion also applies to non-invertible generators, and we provide an explicit proof for $n > 3$ (corresponding to $n > 2$ in their slightly different notation).

# 4 DISCUSSION

## 4.1 HIERARCHY OF INDEPENDENCE

The different independence criteria form a natural hierarchy (see Figure 1). Type D independence is the strongest: it implies all others. Differentiating Type D yields Type $H_2$, and further differentiation gives Type $H_3$, and so on. Type M follows since disjointness is a special case of mutual non-inclusion. Type S is also implied: in the sparsest product-respecting basis, Type D ensures that supports are disjoint, and any linear combination of column vectors from different blocks strictly

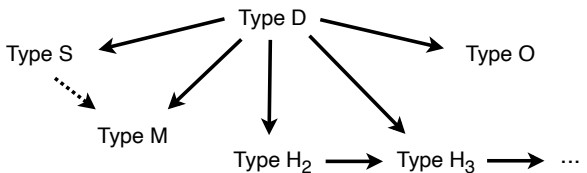

Figure 1: Relations among mechanistic independence types. Arrows indicate logical implications. The dotted arrow holds only in the sparsest product-splitting basis.

enlarges the support, creating a sparsity gap. Finally, Type S implies Type M independence when working in the sparsest product-splitting basis (but not in an arbitrary product-aligned basis).

Since reducibility describes whether a factor can be split into smaller independent subspaces, the implication relations among reducibility types largely mirror those among independence types, except for Type M, which depends on the choice of basis.

This reveals a tradeoff between the identifiability results for Type D and Type S: by enforcing stronger coherence within each factor, we can tolerate stronger interactions between different factors. Relations among the other identifiability results are less direct, since they require additional assumptions (cf. the asymmetric interaction principle of (Brady et al., 2024)).

As with statistical independence, one must distinguish between pairwise and mutual independence. For Types D, M, and $H_n$, the two coincide, but for Type S they differ in general. While mutual independence always implies pairwise independence, Example 1, case $\boldsymbol{B}$, shows a Jacobian where factors are pairwise Type S independent but not mutually so.

## 4.2 Factors of Variation as Connected Graph Components

The factors of variation can also be viewed through graph structures.

**Definition 10** (Graph structures). *Let $\boldsymbol{g}\colon \mathcal{S} \to \mathcal{X}$ be sufficiently smooth, and let $B = (\boldsymbol{u}_1, \ldots, \boldsymbol{u}_{d_s})$ be a basis for $T_{\boldsymbol{s}}\mathcal{S}$. Define the following graphs:*

*(1) $\mathcal{G}^D(\boldsymbol{s}, B) = ([d_s], \mathcal{E}^D)$ with $\mathcal{E}^D = \{(i,j) \in [d_s]^2 \mid D\boldsymbol{g}_{\boldsymbol{s}}(\boldsymbol{u}_i) \bullet D\boldsymbol{g}_{\boldsymbol{s}}(\boldsymbol{u}_j) \neq \boldsymbol{0}\}$*

*(2) $\mathcal{G}^{H_2}(\boldsymbol{s}, B) = ([d_s], \mathcal{E}^{H_2})$ with $\mathcal{E}^{H_2} = \{(i,j) \in [d_s]^2 \mid D^2\boldsymbol{g}_{\boldsymbol{s}}(\boldsymbol{u}_i, \boldsymbol{u}_j) \neq \boldsymbol{0}\}$.*

*(3) $\mathcal{G}^M(\boldsymbol{s}, B) = ([d_s], \mathcal{E}^M)$ with $\mathcal{E}^M = \{(i,j) \in [d_s]^2 \mid \Omega_i \not\perp\!\!\!\perp \Omega_j\}$.*

Consider $\mathcal{G}^D$. In any product-splitting basis, the index sets $\mathcal{C}_i$ and $\mathcal{C}_j$ for $i \neq j$ are disconnected subsets of the vertex set. Type D irreducibility ensures that no $\mathcal{C}_i$ can be further split into disconnected components by using a different basis for $T_{\boldsymbol{s}_i}\mathcal{S}_i$. Thus, the Type D independent and irreducible factors correspond exactly to the connected components of $\mathcal{G}^D$. Moreover, under the assumptions of Type D independence and irreducibility, $\mathcal{G}^D$ cannot have more than $K$ connected components in any basis (see Proposition 5), and in any non-aligned basis it has strictly fewer. Hence, Type D independence and irreducibility could alternatively be characterized by a gap in the number of connected components between aligned and misaligned bases, paralleling the sparsity-gap perspective of Type S.

A similar statement holds for $\mathcal{G}^{H_2}$. If $\boldsymbol{g}$ is second-order separable and satisfies Type $H_2$ independence and irreducibility, then no basis change increases the number of connected components, and any misaligned basis strictly reduces it. For $\mathcal{G}^M$, no analogous conclusion can be drawn, since its definition depends on a specific basis. Nevertheless, the identification of factor subspaces with connected components still applies, though only in the standard basis of $\mathbb{R}^{d_s}$.

This graph-based perspective also connects to recent work on identifiability for local (Euclidean) isometries (Horan et al., 2021), conformal maps, and orthogonal coordinate transformations (Gresele et al., 2021; Buchholz et al., 2022; Ghosh et al., 2023). Each of these function classes can be characterized in terms of their Jacobians: the columns of the Jacobian are mutually orthogonal,

differing only in whether they have unit norm, equal norm, or arbitrary norms. By analogy with Type D independence, we may define *Type O independence* through *orthogonality* in the inner-product sense:

$$\forall i \neq j : \quad D_i \boldsymbol{g_s}(\boldsymbol{u}) \cdot D_j \boldsymbol{g_s}(\boldsymbol{v}) = \boldsymbol{0}.$$

Constructing a graph analogous to $\mathcal{G}^D$, but replacing the Hadamard product with the inner product, yields totally disconnected graphs for these maps when the source factors are one-dimensional.

However, without additional statistical assumptions, identifiability remains limited: even in the smallest class (local isometries), it holds only up to affine transformations. Therefore, to achieve the stronger notion of identifiability pursued in this paper, extra assumptions on the latent distribution are required, even for one-dimensional factors. Nevertheless, such graph constructions may provide a useful tool when combining mechanistic and stochastic independence to recover multidimensional factors.

### 4.3 APPLICABILITY AND LIMITATIONS OF MECHANISTIC INDEPENDENCE

We illustrate the requirements for Type D/M/S/$H_n$ mechanistic independence in the context of image data. Assume that individual latent factors $\boldsymbol{s}_i$ encode distinct objects in a scene (e.g., position, shape, color), and let $\boldsymbol{g}$ denote the rendering process.

Type D independence fails whenever two latent factors influence the same pixel. This excludes shadows, reflections, transparency, and partial occlusions. Type $H_2$ independence fails when the generator cannot be decomposed additively, i.e., when $\boldsymbol{g}(\boldsymbol{s}) \neq \sum_{i \in [K]} \boldsymbol{g}^{(i)}(\boldsymbol{s}_i)$ for any set of functions $\boldsymbol{g}^{(i)}$. Although this assumption is strictly weaker than Type D independence, it still generally disallows partial occlusions, shadows, and reflections. In principle, it permits semi-transparency, but only in the absence of refraction and only when colors mix exactly additively. This condition is further weakened for $n > 2$, but in practice, the calculation of higher-order derivatives is not feasible.

Type M independence fails when the set of pixels affected by a latent coordinate in one group is strictly contained in the set affected by a latent coordinate in another group; for example, when an object is visible solely through its reflection.

Type S independence is more subtle. For one-dimensional slots (i.e., when each object is parameterized by a single latent variable), it can fail only when the fraction of shared affected pixels across slots exceeds one half (lower bound). As already mentioned, it is difficult to convey a similarly strong intuition for multidimensional slots.

## 5 EXPERIMENTS

In an experiment mirroring Brady et al. (2023), we investigated whether the *compositional contrast*

$$C_{\text{comp}}(\hat{\boldsymbol{g}}, \boldsymbol{z}) = \sum_{n=1}^{d_x} \sum_{i=1}^{K} \sum_{j=i+1}^{K} \left\| \frac{\partial \hat{g}_n}{\partial \boldsymbol{z}_i}(\boldsymbol{z}) \right\| \left\| \frac{\partial \hat{g}_n}{\partial \boldsymbol{z}_j}(\boldsymbol{z}) \right\|$$

can serve as an effective surrogate loss for enforcing Type S independence. This question is motivated by the observation that some generators have latent components that are Type S independent but not Type D independent, yet minimizing $C_{\text{comp}}$ can nonetheless enforce Type S independence in the learned representation (see Example 2). As argued in Section 3.3, Type S independence is likely to hold when only a small number of observation dimensions are influenced by multiple latent factors (slots).

To examine this, we generate synthetic datasets with varying degrees of overlap between the sets of observation dimensions affected by different slots, as illustrated on the right in Figure 2. Latent variables are sampled from a standard normal distribution, and observations are produced by passing them through an invertible MLP whose Jacobian is constructed to have the desired support structure. Only when the overlap is $0\%$ does the generator satisfy Type D independence.

We train an autoencoder with reconstruction loss and compositional contrast, $\mathcal{L} = \mathcal{L}_{\text{recon}} + \lambda C_{\text{comp}}$, across five random seeds, using $L = K \in \{2, 3, 5\}$ slots and regularization strengths $\lambda \in \{10^{-2}, 1\}$. For comparability across hyperparameters, we normalize $C_{\text{comp}}$ (see Appendix D for details).

Figure 2 indicates that, for sufficiently small overlaps, $C_{\text{comp}}$ acts as a reliable proxy for Type S independence. However, as the overlap ratio increases, the likelihood of convergence to bad local minima also grows. Identifying more robust surrogate losses remains an open challenge, which we leave for future work. Further experiments can be found in Appendix D.

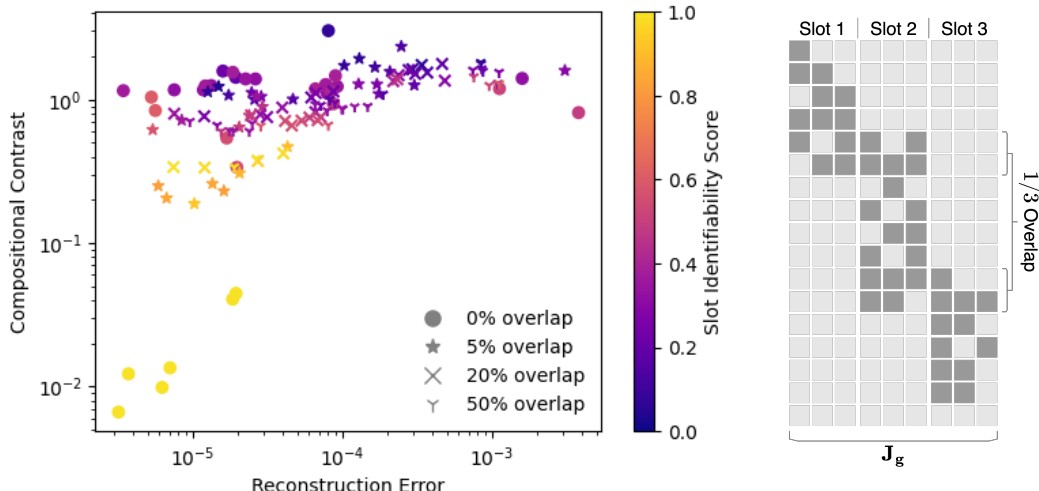

Figure 2: Slot Identifiability Score (SIS) over reconstruction loss and compositional contrast for different support overlaps.

## 6 RELATED WORK

Beyond the already mentioned approaches, a number of other works establish identifiability by imposing structural constraints. Moran et al. (2021) prove identifiability in sparse VAEs by enforcing sparsity in the decoder; while our framework does not subsume theirs, their synthetic dataset can also be shown to satisfy Theorems 3 and 4. Rhodes & Lee (2021) provide empirical evidence that penalizing the decoder Jacobian with an $\ell_1$-norm helps break rotational symmetries in VAEs – our results can be seen as offering the corresponding theoretical justification. In contrast, Lachapelle et al. (2022) obtain identifiability of latent factors by enforcing sparsity on causal mechanisms, while Reizinger et al. (2023) connect sparsity patterns in the Jacobian to identifiable causal graphs in nonlinear ICA.

A distinctive aspect of our work is that we establish identifiability at the subspace level, whereas most prior results assume that each latent factor is captured in a single dimension. Recent research has also examined block-identifiability of latent variables under paired observations. These include content–style separation via data augmentation (Von Kügelgen et al., 2021) or multiple views (Daunhawer et al., 2023), block-disentanglement under sparse perturbations (Fumero et al., 2021; Ahuja et al., 2022; Mansouri et al., 2023), and temporal formulations leveraging causal graphs (Lachapelle & Lacoste-Julien, 2022; Lachapelle et al., 2024).

## 7 CONCLUSION

In this work, we have developed a unifying framework for disentanglement and identifiability based on *mechanistic independence*. By formulating independence at the level of generative mechanisms rather than distributions, we obtained identifiability results for subspaces that hold under minimal assumptions on the latent density and extend to nonlinear, non-invertible generators. Our analysis revealed a hierarchy of independence criteria , and clarified how these conditions trade off internal factor coherence against cross-factor interactions. Overall, the results establish when disentangled representations are identifiable without relying on statistical assumptions, providing a theoretical foundation for future work that explores other mechanistic independence criteria or combines mechanistic and stochastic assumptions.

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

## NOTATION INDEX

$a$    A scalar

$\boldsymbol{a}$    A vector

$\boldsymbol{A}$    A matrix

$\mathcal{A}$    A set

$a_i$    $i$-th coordinate of $\boldsymbol{a}$ (index starting at 1)

$\boldsymbol{a}_i$    $i$-th factor of $\boldsymbol{a}$ if $\boldsymbol{a}$ lives in a product space

$a_{ij}$    $j$-th coordinate of the $i$-th factor of $\boldsymbol{a}$

$d_x$    Dimensionality of observations

$d_s$    Dimensionality of ground-truth latents

$d_i$    Dimensionality of the $i$-th latent factor

$d_z$    Dimensionality of the learned representation

$\boldsymbol{D}_n$    Duplication matrix for $n \times n$ matrices

$D\boldsymbol{g_s}$    Differential of $\boldsymbol{g}$ at $\boldsymbol{s}$

$D_i\boldsymbol{g_s}$    Partial derivative w.r.t. the $i$-th factor $D\boldsymbol{g_s} \circ \iota_i$

$D^3_{i,j}\boldsymbol{g_s}$    Mixed derivative $D^3\boldsymbol{g_s} \circ (\iota_i, \iota_j, \mathrm{id})$

$\boldsymbol{e}_i$    Standard basis vector with a 1 at position $i$

$\boldsymbol{f}$    Ground-truth encoder

$\hat{\boldsymbol{f}}$    Learned encoder

$\boldsymbol{g}$    Ground-truth decoder

$\hat{\boldsymbol{g}}$    Learned decoder

$\mathcal{G} = (\mathcal{V}, \mathcal{E})$    A graph $\mathcal{G}$ defined by a set of vertices $\mathcal{V}$ and edges $\mathcal{E}$

$\boldsymbol{h}$    Mapping from ground-truth to learned latents

$\boldsymbol{I}$    Identity matrix with implied size from context

$\boldsymbol{I}_n$    Identity matrix of size $n \times n$

$\boldsymbol{J_f}$    Jacobian matrix of $\boldsymbol{f} : \mathbb{R}^n \to \mathbb{R}^m$ ($\boldsymbol{J_f} \in \mathbb{R}^{m \times n}$)

$\iota_i$    The $i$-th canonical inclusion map

$K$    Number of latent factors

$L$    Number of factors in learned representation

$\boldsymbol{L}_n$    Elimination matrix for $n \times n$ matrices

$\mathbb{P}$    A probability distribution

$\boldsymbol{s}$    Ground-truth latent variable

$\mathcal{S}$    Ground-truth latent space

$\mathcal{S}_i$    $i$-th latent subspace ($\mathcal{S} \subseteq \mathcal{S}_1 \times \cdots \times \mathcal{S}_K$)

$\mathrm{supp}(\cdot)$    Support of a matrix (index set of nonzero elements) or support of a function (set of elements mapped to values not equal to zero)

$T_{\boldsymbol{s}}\mathcal{S}$    Tangent space of $\mathcal{S}$ at $\boldsymbol{s}$

$\boldsymbol{v}$    Mapping from learned to ground-truth latents

$\boldsymbol{x}$    Observation or measurement

$\mathcal{X}$    Data manifold ($\boldsymbol{x} \in \mathcal{X} \subseteq \mathbb{R}^{d_x}$)

$\boldsymbol{z}$    Learned representation (or encoding)

$\mathcal{Z}$    Learned representation space

$\times$    Direct product

$\oplus$    Direct sum

$\bullet$    Hadamard product (element-wise product)

$\otimes$    Kronecker product

$\odot$    Row-wise Kronecker product (also face-splitting product)

$\backslash$    Set subtraction

$\pitchfork$    Mutual non-inclusion ($\mathcal{A} \nsubseteq \mathcal{B} \,\wedge\, \mathcal{A} \nsupseteq \mathcal{B}$)

$|\mathcal{A}|$    Cardinality of set $\mathcal{A}$ (the number of elements in $\mathcal{A}$)

$[n]$    The set $\{1, 2, \ldots, n\}$ for $n \in \mathbb{N}$

$f \circ g$    Composition of the functions $f$ and $g$

$\|\boldsymbol{x}\|_0$    $\ell_0$ norm of $\boldsymbol{x}$

# A   PROOFS

Before we turn to the theorems and proofs, let us recall the following definitions.

**Definition 11** (Local disentanglement). *A decoder $\hat{g}\colon \mathcal{Z} \to \mathcal{X}$ is* locally disentangled *w.r.t. a generator $g\colon \mathcal{S} \to \mathcal{X}$ if for every $s^* \in \mathcal{S}$ and $z^* \in \mathcal{Z}$ with $g(s^*) = \hat{g}(z^*)$ there exist a neighborhood $\mathcal{U} \subseteq \mathcal{S}$ of $s^*$ and a decomposable map $h\colon \mathcal{U} \to \mathcal{Z}$ such that*

$$g\big|_{\mathcal{U}} = \hat{g} \circ h \qquad and \qquad h(s^*) = z^*.$$

**Definition 12** (k-factor slice). *Let $k \in \{0, \ldots, K\}$, and let $\mathcal{I} \subseteq [K]$ be an index set with $|\mathcal{I}| = K - k$. If $\mathcal{S}$ is a subset of the product space $\mathcal{S}_1 \times \cdots \times \mathcal{S}_K$, a $k$-factor* slice *(or simply a $k$-slice) of $\mathcal{S}$ is any set of the form*

$$\mathcal{U} = \{s \in \mathcal{S} \mid s_i = c_i \text{ for all } i \in \mathcal{I}\},$$

*where $c_i \in \mathcal{S}_i$ for $i \in \mathcal{I}$ are fixed constants.*

Put simply, a $k$-slice is a subspace in which all but $k$ factors are held constant.

**Remark 1.** *Path-connectedness of $\mathcal{S} \subseteq \prod_{i=1}^{K} \mathcal{S}_i$ and path-connectedness of its $(K-1)$-slices are related but independent properties: neither one implies the other (see Figure 3). More generally, for $K > 2$, connectedness of 1-slices and 2-slices are likewise independent (for $K = 2$ they coincide trivially). A further related notion is* orthogonal convexity*, which can be interpreted as the property that all 1-slices are path-connected (when each factor is one-dimensional).*

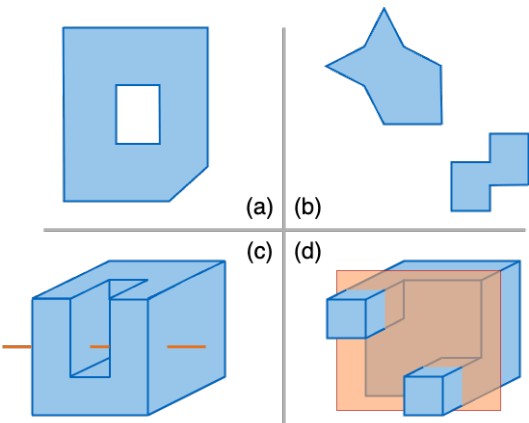

Figure 3: Examples illustrating independence of slice- and set-level connectedness. (a) $\mathcal{S}$ is path-connected, but not every 1-slice is connected. (b) $\mathcal{S}$ is not path-connected, though every 1-slice is connected. (c) Some 1-slices are disconnected, but every 2-slice is connected. (d) Some 2-slices are disconnected, but every 1-slice is connected.

## A.1   PROOF OF THEOREM 1

**Theorem 1** (Global identifiability). *Let $\mathcal{S}$ be an open subspace of the product manifold $\prod_{i=1}^{K} \mathcal{S}_i$, where each factor $\mathcal{S}_i$ has positive dimension. Assume $\hat{g} : \mathcal{Z} \to \mathcal{X}$ is locally disentangled w.r.t. $g : \mathcal{S} \to \mathcal{X}$. Then $\hat{g}$ is globally disentanglement if:*

    *(1) $\mathcal{S}$ is simply connected.*

    *(2) Every $(K$-$1)$-slice of $\mathcal{S}$ is path-connected.*

    *(3) $g\colon \mathcal{S} \to \mathcal{X}$ is continuous and locally injective.*

    *(4) $\hat{g}\colon \mathcal{Z} \to \mathcal{X}$ is a covering map.*

*Proof.* By the covering-space lifting criterion, the conditions that $\mathcal{S}$ is simply connected and $\hat{g}$ a covering map imply the existence of a continuous lift[8] $h : \mathcal{S} \to \mathcal{Z}$ satisfying

$$g = \hat{g} \circ h.$$

This lift is locally injective since $g$ is locally injective.

We first note that local disentanglement induces a local decomposable structure in $h$ and that the association $\sigma \colon [K] \to [K]$ between source and target factors is uniquely determined when restricted to the global lift. To show this, fix some $s^* \in \mathcal{S}$. By local disentanglement, applied with

$$z^* = h(s^*),$$

there exists a neighborhood $\mathcal{U}$ of $s^*$ and a decomposable map $h^{\mathcal{U}} : \mathcal{U} \to \mathcal{Z}$ such that

$$\hat{g} \circ h^{\mathcal{U}} = g\big|_{\mathcal{U}}$$

and

$$h^{\mathcal{U}}(s^*) = h(s^*).$$

Shrinking $\mathcal{U}$ if necessary, we may assume that $\mathcal{U}$ is connected. Since both $h^{\mathcal{U}}$ and $h\big|_{\mathcal{U}}$ are lifts of $g\big|_{\mathcal{U}}$ through the local homeomorphism $\hat{g}$ (implied by the covering map condition), and since they agree at $s^*$, uniqueness of lifts on a connected set gives

$$h^{\mathcal{U}} = h\big|_{\mathcal{U}}.$$

Hence the global lift $h$ is locally decomposable.

Let $\sigma : [K] \to [K]$ be the permutation with respect to which $h\big|_{\mathcal{U}}$ is decomposable. We claim that every local decomposable description of $h$ uses the same $\sigma$. Let $s \in \mathcal{S}$, and choose a path $\gamma : [0,1] \to \mathcal{S}$ with

$$\gamma(0) = s^*, \qquad \gamma(1) = s.$$

Cover $\gamma([0,1])$ by connected open sets $\mathcal{U}_\alpha$ with $\alpha \in [0,1]$ such that $h\big|_{\mathcal{U}_\alpha}$ is decomposable. By compactness, there exists a finite subcover $\mathcal{U}_1, \ldots, \mathcal{U}_M$. Using the Lebesgue number lemma, choose a partition

$$0 = \alpha_0 < \alpha_1 < \cdots < \alpha_M = 1 \quad \text{such that} \quad \gamma([\alpha_{m-1}, \alpha_m]) \subset \mathcal{U}_m \text{ for each } m \in [M].$$

Then $\gamma(\alpha_m) \in \mathcal{U}_m \cap \mathcal{U}_{m+1}$, so consecutive sets intersect.

Suppose $h\big|_{\mathcal{U}_m}$ is decomposable with respect to $\sigma_m$, and $h\big|_{\mathcal{U}_{m+1}}$ is decomposable with respect to $\sigma_{m+1}$. On the open set $\mathcal{U}_m \cap \mathcal{U}_{m+1}$, both descriptions describe the same map $h$. We show that

$$\sigma_m = \sigma_{m+1}.$$

Fix some $i \in [K]$ and suppose, for contradiction, that

$$\sigma_m(i) \neq \sigma_{m+1}(i).$$

Choose a point $p \in \mathcal{U}_m \cap \mathcal{U}_{m+1}$. Since $\mathcal{S}$ is open in the product and $\mathcal{S}_i$ has positive dimension, there is a nonconstant path

$$\eta : (-\epsilon, \epsilon) \to \mathcal{U}_m \cap \mathcal{U}_{m+1}$$

for sufficiently small $\epsilon$ with $\eta(0) = p$ such that only the $i$-th source coordinate varies along $\eta$.

Because $h$ is decomposable with respect to $\sigma_m$ on $\mathcal{U}_m$, all target components except possibly the $\sigma_m(i)$-th one are constant along $\eta$. Because $h$ is decomposable with respect to $\sigma_{m+1}$ on $\mathcal{U}_{m+1}$, all target components except possibly the $\sigma_{m+1}(i)$-th one are constant along $\eta$. Since

$$\sigma_m(i) \neq \sigma_{m+1}(i),$$

every target component is constant along $\eta$. Hence $h \circ \eta$ is constant. This contradicts local injectivity of $h$. Therefore

$$\sigma_m(i) = \sigma_{m+1}(i).$$

---

[8]Given maps $f : \mathcal{A} \to \mathcal{X}$ and $g : \mathcal{B} \to \mathcal{X}$, a lift of $f$ to $\mathcal{B}$ is a map $h : \mathcal{A} \to \mathcal{B}$ such that $f = g \circ h$.

Since $i$ was arbitrary,

$$\sigma_m = \sigma_{m+1}.$$

By induction along the chain, every local decomposable description of $\boldsymbol{h}$ along $\boldsymbol{\gamma}$ uses the same permutation $\sigma$. Since $\boldsymbol{s}$ was arbitrary and $\mathcal{S}$ is path-connected, the same $\sigma$ applies locally everywhere on $\mathcal{S}$.

It remains to show that $\boldsymbol{h}$ is globally decomposable with respect to this $\sigma$. Write

$$\boldsymbol{h} = (\boldsymbol{h}_1, \ldots, \boldsymbol{h}_K).$$

Fix some $j \in [K]$, and let $\boldsymbol{s}, \boldsymbol{s}' \in \mathcal{S}$ satisfy

$$\boldsymbol{s}_{\sigma^{-1}(i)} = \boldsymbol{s}'_{\sigma^{-1}(i)}.$$

By assumption, $\boldsymbol{s}$ and $\boldsymbol{s}'$ lie in the same path-connected $(K\text{-}1)$-slice. Now choose a path $\boldsymbol{\gamma} : [0,1] \to \mathcal{S}$ inside this slice that connects

$$\boldsymbol{s} = \boldsymbol{\gamma}(0) \qquad \text{with} \qquad \boldsymbol{s}' = \boldsymbol{\gamma}(1).$$

For every $t \in [0,1]$, local decomposability gives an open neighborhood $\mathcal{U}_t$ of $\boldsymbol{\gamma}(t)$ on which $\boldsymbol{h}$ agrees with a decomposable local lift with respect to $\sigma$. The sets $\mathcal{U}_t$ form an open cover of $\boldsymbol{\gamma}([0,1])$. Following the same argument as before, we can choose a partition

$$0 = t_0 < t_1 < \cdots < t_M = 1$$

such that for each $m$,

$$\boldsymbol{\gamma}([t_{m-1}, t_m]) \subseteq \mathcal{U}_m$$

for some such open set $\mathcal{U}_m$.

On each $\mathcal{U}_m$, the component $\boldsymbol{h}_j$ depends only on the coordinate indexed by $\sigma^{-1}(j)$, which is constant along $\boldsymbol{\gamma}$. Hence $\boldsymbol{h}_j \circ \boldsymbol{\gamma}$ is constant on each interval $[t_{m-1}, t_m]$. Since consecutive intervals meet, the constants agree. Therefore

$$\boldsymbol{h}_j(\boldsymbol{s}) = \boldsymbol{h}_j(\boldsymbol{\gamma}(0)) = \boldsymbol{h}_j(\boldsymbol{\gamma}(1)) = \boldsymbol{h}_j(\boldsymbol{s}').$$

Thus $\boldsymbol{h}_j$ depends only on $\boldsymbol{s}_{\sigma^{-1}(j)}$. Since $j$ was arbitrary, $\boldsymbol{h}$ is decomposable with respect to $\sigma$. So $\hat{\boldsymbol{g}}$ is globally disentangled with respect to $\boldsymbol{g}$. $\square$

**Remark 2** (Possible relaxations of Theorem 1). *The assumptions that $\mathcal{S}$ is simply connected and $\hat{\boldsymbol{g}}$ is a covering map are only used once at the beginning of the proof to infer that there exists a global continuous lift. The assumption that $\mathcal{S}$ is simply connected is mainly a convenient way to rule out global "branch switching" of local lifts. It can be weakened.*

*First, it is enough to assume that $\mathcal{S}$ is path-connected and that the usual lifting condition holds. Namely, fix basepoints*

$$\boldsymbol{s}^* \in \mathcal{S}, \qquad \boldsymbol{z}^* \in \mathcal{Z}, \qquad \boldsymbol{g}(\boldsymbol{s}^*) = \hat{\boldsymbol{g}}(\boldsymbol{z}^*).$$

*Then the simply connectedness assumption may be replaced by*

$$(\boldsymbol{g})_* \pi_1(\mathcal{S}, \boldsymbol{s}^*) \subseteq (\hat{\boldsymbol{g}})_* \pi_1(\mathcal{Z}, \boldsymbol{z}^*).$$

*Here $\pi_1(\mathcal{S}, \boldsymbol{s}^*)$ denotes the fundamental group of $\mathcal{S}$ based at $\boldsymbol{s}^*$, whose elements are loops in $\mathcal{S}$ starting and ending at $\boldsymbol{s}^*$, considered up to continuous deformation. The map $(\boldsymbol{g})_*$ sends such a loop in $\mathcal{S}$ to the corresponding loop in $\mathcal{X}$ obtained by applying $\boldsymbol{g}$. Similarly, $(\hat{\boldsymbol{g}})_*$ records which loops in $\mathcal{X}$ arise as images under $\hat{\boldsymbol{g}}$ of loops in $\mathcal{Z}$ based at $\boldsymbol{z}^*$. Thus the condition says that every loop in $\mathcal{X}$ produced by moving around in $\mathcal{S}$ through $\boldsymbol{g}$ can be lifted to a loop in $\mathcal{Z}$ through $\hat{\boldsymbol{g}}$. Equivalently, following a local lift of $\boldsymbol{g}$ around any loop in $\mathcal{S}$ returns to the same branch of the lift.*

*Second, the assumption that $\hat{\boldsymbol{g}}$ is a covering map may itself be derived from more elementary regularity assumptions. Suppose $\mathcal{X}, \mathcal{Z}$ are manifolds, $\mathcal{X}$ is connected, and $\hat{\boldsymbol{g}} : \mathcal{Z} \to \mathcal{X}$ is a surjective local homeomorphism. Then $\hat{\boldsymbol{g}}$ is a covering map under either of the following additional assumptions:*

*(1) $\hat{\boldsymbol{g}}$ is proper, or*

*(2) $\left| \hat{\boldsymbol{g}}^{-1}(\boldsymbol{x}) \right| = n < \infty$ for all $\boldsymbol{x} \in \mathcal{X}$.*

**Remark 3.** *To extend local partial disentanglement to global partial disentanglement (with $L < K$), condition (2) in Theorem 1 must be replaced by the following: for every $j \in [L]$, each slice*

$$\left\{ \boldsymbol{s} \in \mathcal{S} : \boldsymbol{s}_i = \bar{\boldsymbol{s}}_i \text{ for all } i \in \sigma^{-1}(j) \right\}$$

*is path-connected. Here, $\sigma$ may be chosen from any locally decomposable map.*

A.2 PROOF OF PROPOSITION 1

**Lemma 1.** *Let $\mathcal{S} \subseteq \prod_{i=1}^{K} \mathcal{S}_i, \mathcal{Z} \subseteq \prod_{j=1}^{L} \mathcal{Z}_j$, and suppose $g \colon \mathcal{S} \to \mathcal{X}$ and $\hat{g} \colon \mathcal{Z} \to \mathcal{X}$ are local homeomorphisms. Assume that for every $s^* \in \mathcal{S}$ there exists $z^* \in \mathcal{Z}$ with $g(s^*) = \hat{g}(z^*)$. Moreover, suppose that for each such $z^*$ there exist*

- *a neighborhood $\mathcal{U} \subseteq \mathcal{Z}$ of $z^*$,*

- *a surjection $\sigma \colon [K] \to [L]$, and*

- *maps $v_i \colon \mathcal{Z}_{\sigma(i)} \to \mathcal{S}_i$ for $i \in [K]$,*

*such that for all $z \in \mathcal{U}$,*

$$\hat{g}(z) = g\big(v_1(z_{\sigma(1)}), \dots, v_K(z_{\sigma(K)})\big). \tag{9}$$

*Then $\hat{g}$ is locally disentangled with respect to $g$.*

*Proof.* Fix an arbitrary $s^* \in \mathcal{S}$ and pick $z^* \in \mathcal{Z}$ with $\hat{g}(z^*) = g(s^*)$. By hypothesis at $z^*$, there is a neighborhood $\mathcal{U} = \prod_{j=1}^{L} \mathcal{U}_j \subseteq \mathcal{Z}$, a surjection $\sigma$, and maps $v_i$ giving Equation 9 on $\mathcal{U}$.

Shrink to a neighborhood $\mathcal{W} \subseteq \mathcal{S}$ of $s^*$ on which $g \colon \mathcal{W} \to g(\mathcal{W})$ is a homeomorphism, and shrink $\mathcal{U}$ if necessary so that $\hat{g}(\mathcal{U}) \subseteq g(\mathcal{W})$. Define

$$\psi := g^{-1} \circ \hat{g} : \mathcal{U} \longrightarrow \mathcal{W}.$$

Then $\psi$ is a homeomorphism onto its image with $\psi(z^*) = s^*$.

For each $j \in [L]$ set

$$\phi_j : \mathcal{U}_j \longrightarrow \prod_{i \in \sigma^{-1}(j)} \mathcal{S}_i, \qquad \phi_j(\alpha) := \big(v_i(\alpha)\big)_{i \in \sigma^{-1}(j)}.$$

Then for $z \in \mathcal{U}$ Equation 9 is equivalent to

$$\varrho_\sigma\big(\psi(z)\big) = \big(\phi_1(z_1), \dots, \phi_L(z_L)\big), \tag{10}$$

where $\varrho_\sigma$ is a reindexing homeomorphism $s \mapsto \big((s_i)_{i \in \sigma^{-1}(j)}\big)_{j=1}^{L}$. Therefore, each $\phi_i$ must be injective, because the left hand side of Equation 10 is a homeomorphism onto its image.

Since $\psi(\mathcal{U})$ is an open neighborhood of $s^*$ in the product space $\prod_i \mathcal{S}_i$, we can choose product neighborhoods $\mathcal{V}_i \subseteq \mathcal{S}_i$ with

$$\prod_{i=1}^{K} \mathcal{V}_i \subseteq \psi(\mathcal{U}).$$

Then for each $j$ we have $\prod_{i \in \sigma^{-1}(j)} \mathcal{V}_i \subseteq \phi_j(\mathcal{U}_j)$, and we set

$$h_j := \phi_j^{-1}\big|_{\prod_{i \in \sigma^{-1}(j)} \mathcal{V}_i} : \prod_{i \in \sigma^{-1}(j)} \mathcal{V}_i \longrightarrow \mathcal{U}_j.$$

Finally, for any $s \in \prod_i \mathcal{V}_i$, define $z := \big(h_j((s_i)_{i \in \sigma^{-1}(j)})\big)_{j=1}^{L}$. Then, by construction and Equation 10, $\psi^{-1}(s) = z$, hence

$$g(s) = \hat{g}(z) = \hat{g}\Big(h_1\big((s_i)_{i \in \sigma^{-1}(1)}\big), \dots, h_L\big((s_i)_{i \in \sigma^{-1}(L)}\big)\Big).$$

Therefore, $\hat{g}$ is locally disentangled with respect to $g$ on a neighborhood of the arbitrary point $s^* \in \mathcal{S}$. □

**Proposition 1.** *Let $\mathcal{F}$ be a class of surjective local homeomorphisms $g \colon \mathcal{S}^g \to \mathcal{X}$ into the same observation space, where each $\mathcal{S}^g$ is an open subset of a product space. Define $\hat{g} \sim_{ld} g$ to mean that $\hat{g}$ is locally disentangled w.r.t. $g$. Suppose that*

$$g \sim_{ld} g \qquad \text{for all } g \in \mathcal{F}.$$

*Then $\sim_{ld}$ is an equivalence relation on $\mathcal{F}$.*

*Proof.* By assumption, $\sim_{ld}$ is reflexive.

We next prove transitivity. Suppose $\hat{g} \sim_{ld} g$ and $g \sim_{ld} \bar{g}$. Let $y \in \mathcal{S}^g$ and $z \in \mathcal{S}^{\hat{g}}$ satisfy

$$\bar{g}(y) = \hat{g}(z).$$

Since $g$ is surjective into the same observation space, choose $s \in \mathcal{S}^g$ such that

$$g(s) = \bar{g}(y) = \hat{g}(z).$$

From $g \sim_{ld} \bar{g}$, it follows that there exist an open neighborhood $\mathcal{U}$ of $y$ and a decomposable map $h : \mathcal{U} \to \mathcal{S}^g$ such that

$$\bar{g}\big|_{\mathcal{U}} = g \circ h, \qquad h(y) = s.$$

Analogously from $\hat{g} \sim_{ld} g$, there exist an open neighborhood $\mathcal{V}$ of $s$ and a decomposable map $\eta : \mathcal{V} \to \mathcal{S}^{\hat{g}}$ such that

$$g\big|_{\mathcal{V}} = \hat{g} \circ \eta, \qquad \eta(s) = z.$$

After shrinking $\mathcal{U}$, we may assume $h(\mathcal{U}) \subseteq \mathcal{V}$. Then $\eta \circ h$ is decomposable as the composition of decomposable maps, and

$$\bar{g}\big|_{\mathcal{U}} = g \circ h = \hat{g} \circ \eta \circ h, \qquad (\eta \circ h)(y) = z.$$

Hence $\hat{g} \sim_{ld} \bar{g}$. Thus $\sim_{ld}$ is transitive.

Symmetry follows from Lemma 1. Thus $\sim_{ld}$ is reflexive, transitive, and symmetric, and hence an equivalence relation on $\mathcal{F}$. □

**Proposition 2.** *Let $\mathcal{F}$ be a class of homeomorphisms $g : \mathcal{S}^g \to \mathcal{X}$ into the same observation space, where each $\mathcal{S}^g \subseteq \prod_{i=1}^{K} \mathcal{S}_i^g$ is open. Assume that every $(K-1)$-slice of each $\mathcal{S}^g$ is path-connected. Define $\hat{g} \sim_d g$ to mean that $\hat{g}$ is disentangled w.r.t. $g$. Then $\sim_d$ is an equivalence relation on $\mathcal{F}$.*

*Proof.* Reflexivity follows immediately, because the identity map is decomposable. Transitivity follows from the fact that the composition of decomposable maps is decomposable. It remains to prove symmetry.

Let $g : \mathcal{S} \to \mathcal{X}$ and $\hat{g} : \mathcal{Z} \to \mathcal{X}$. Suppose $\hat{g} \sim_d g$, and set

$$h := \hat{g}^{-1} \circ g.$$

Then $h : \mathcal{S} \to \mathcal{Z}$ is a homeomorphism and is decomposable. Thus, for some permutation $\sigma$,

$$h_i(s) = h^{(i)}(s_{\sigma^{-1}(i)}), \qquad i \in [K].$$

We show that $h^{-1}$ is also decomposable.

Fix $i \in [K]$, and write $j = \sigma^{-1}(i)$. We first observe that $h^{(i)}$ is locally injective on the coordinate values that occur in $\mathcal{S}$. If $h^{(i)}$ were not locally injective at some coordinate value $s_j$, then one could vary only the $j$-th coordinate in a sufficiently small product neighborhood while leaving all other coordinates fixed. This would produce two distinct points in $\mathcal{S}$ with the same image under $h$, contradicting the fact that $h$ is a homeomorphism.

Now consider two points $z, \tilde{z} \in \mathcal{Z}$ that lie in the same $(K-1)$-slice obtained by fixing the $i$-th coordinate, that is,

$$z_i = \tilde{z}_i.$$

By assumption, this slice is path-connected, so there exists a continuous path $\gamma : [0,1] \to \mathcal{Z}$ such that

$$\gamma(0) = z, \qquad \gamma(1) = \tilde{z}, \qquad \gamma_i(t) = z_i \quad \text{for all } t \in [0,1].$$

Let

$$\eta(t) := h^{-1}(\gamma(t)).$$

Then $\eta$ is continuous, and for all $t$,

$$h^{(i)}(\eta_j(t)) = h_i(\eta(t)) = \gamma_i(t) = z_i.$$

Since $\boldsymbol{h}^{(i)}$ is locally injective, the continuous function $\eta_j$ is locally constant, and because $[0, 1]$ is connected, it is constant. Hence $\eta_j(0) = \eta_j(1)$ or equivalently,

$$\left(\boldsymbol{h}^{-1}(\boldsymbol{z})\right)_j = \left(\boldsymbol{h}^{-1}(\tilde{\boldsymbol{z}})\right)_j.$$

Thus, the $j$-th coordinate of $\boldsymbol{h}^{-1}(\boldsymbol{z})$ depends only on the $i$-th coordinate of $\boldsymbol{z}$, where $i = \sigma(j)$.

Since this holds for every $j$, there exist maps $\tilde{\boldsymbol{h}}^{(j)} \colon \mathcal{Z}_{\sigma(j)} \to \mathcal{S}_j$ such that

$$\boldsymbol{h}^{-1}(\boldsymbol{z}) = \left(\tilde{\boldsymbol{h}}^{(1)}(\boldsymbol{z}_{\sigma(1)}), \ldots, \tilde{\boldsymbol{h}}^{(K)}(\boldsymbol{z}_{\sigma(K)})\right).$$

Therefore $\boldsymbol{h}^{-1}$ is decomposable, with permutation $\sigma^{-1}$. Since $\hat{\boldsymbol{g}} = \boldsymbol{g} \circ \boldsymbol{h}^{-1}$, we conclude that $\boldsymbol{g} \sim_d \hat{\boldsymbol{g}}$. Hence the relation is symmetric, and so $\sim_d$ is an equivalence relation. □

### A.3 PROOF OF THEOREM 2

**Definition 13** (Mechanistic independence of Type D). *We say that $\mathcal{S}_i$ and $\mathcal{S}_j$ (equivalently, $\boldsymbol{s}_i$ and $\boldsymbol{s}_j$) are* mechanistically independent of Type D *if, for all $\boldsymbol{s} \in \mathcal{S}$, $\boldsymbol{\xi} \in T_{\boldsymbol{s}_i}\mathcal{S}_i$, and $\boldsymbol{\eta} \in T_{\boldsymbol{s}_j}\mathcal{S}_j$,*

$$D_i\boldsymbol{g_s}(\boldsymbol{\xi}) \bullet D_j\boldsymbol{g_s}(\boldsymbol{\eta}) = \boldsymbol{0}, \tag{11}$$

*where $\bullet$ denotes the element-wise (Hadamard) product in $\mathbb{R}^{d_x}$.*

Independence of the $\mathcal{Z}_j$ is analogously defined based on $\hat{\boldsymbol{g}}$.

**Definition 14** (Reducibility of Type D). *We say that $\mathcal{S}_i$ is* reducible of Type D *if there exists $\boldsymbol{s} \in \mathcal{S}$ such that $T_{\boldsymbol{s}_i}\mathcal{S}_i$ admits a nontrivial direct-sum decomposition $T_{\boldsymbol{s}_i}\mathcal{S}_i = U \oplus V$ with the property that, for all $\boldsymbol{\xi} \in U$ and $\boldsymbol{\eta} \in V$,*

$$D_i\boldsymbol{g_s}(\boldsymbol{\xi}) \bullet D_i\boldsymbol{g_s}(\boldsymbol{\eta}) = \boldsymbol{0}. \tag{12}$$

*If no such decomposition exists, we call $\mathcal{S}_i$* irreducible of Type D.

**Lemma 2.** *Let $\{\mathcal{Z}_j\}_{j=1}^L$ and $\{\mathcal{S}_i\}_{i=1}^K$ be smooth manifolds of positive dimension with $L \leq K$, and let $\mathcal{Z} \subseteq \mathcal{Z}_1 \times \cdots \times \mathcal{Z}_L$ and $\mathcal{S} \subseteq \mathcal{S}_1 \times \cdots \times \mathcal{S}_K$ be open subsets. Suppose $\boldsymbol{v} \colon \mathcal{Z} \to \mathcal{S}$ is a diffeomorphism such that for every $\boldsymbol{z} \in \mathcal{Z}$ there exists a surjection $\sigma_{\boldsymbol{z}} \colon [K] \to [L]$ satisfying*

$$D_j(\boldsymbol{\pi}_i \circ \boldsymbol{v})_{\boldsymbol{z}} = 0, \quad \text{for all } i \in [K], \ j \neq \sigma_{\boldsymbol{z}}(i),$$

*where $\boldsymbol{\pi}_i \colon \mathcal{S} \to \mathcal{S}_i$ denotes a canonical projection. Then for every $\boldsymbol{z} \in \mathcal{Z}$ there exists a neighborhood $\mathcal{U}$ of $\boldsymbol{z}$ such that $\sigma_{\boldsymbol{z}'} = \sigma_{\boldsymbol{z}}$ for all $\boldsymbol{z}' \in \mathcal{U}$, and moreover $\boldsymbol{v}_i(\boldsymbol{z}')$ depends only on the component $\boldsymbol{z}'_{\sigma(i)}$ for each $i \in [K]$.*

*Proof.* At each $\boldsymbol{z} \in \mathcal{Z}$, the differential $D\boldsymbol{v}_{\boldsymbol{z}}$ has block form

$$D\boldsymbol{v}_{\boldsymbol{z}} = \bigoplus_{j=1}^L \Phi_{\boldsymbol{z},j}, \qquad \Phi_{\boldsymbol{z},j} \colon T_{\boldsymbol{z}_j}\mathcal{Z}_j \to \bigoplus_{i \in \sigma_{\boldsymbol{z}}^{-1}(j)} T_{\boldsymbol{\pi}_i(\boldsymbol{v}(\boldsymbol{z}))}\mathcal{S}_i.$$

Since $\boldsymbol{v}$ is a diffeomorphism, $D\boldsymbol{v}_{\boldsymbol{z}}$ is an isomorphism. Hence each block $\Phi_{\boldsymbol{z},j}$ must also be an isomorphism, and in particular

$$\dim(\mathcal{Z}_j) = \sum_{i \in \sigma_{\boldsymbol{z}}^{-1}(j)} \dim(\mathcal{S}_i).$$

The maps $\boldsymbol{z} \mapsto D_j(\boldsymbol{\pi}_i \circ \boldsymbol{v})_{\boldsymbol{z}}$ vary smoothly with $\boldsymbol{z}$. Thus, if $\Phi_{\boldsymbol{z},j}$ is an isomorphism at $\boldsymbol{z}$, it remains so in a neighborhood of $\boldsymbol{z}$, since invertibility is an open condition. This implies $\sigma_{\boldsymbol{z}'}^{-1}(j) \supseteq \sigma_{\boldsymbol{z}}^{-1}(j)$ for all $j \in [L]$ as we assumed the $\mathcal{S}_i$ have positive dimension. Because each $\sigma_{\boldsymbol{z}'}$ is surjective, we must have $\sigma_{\boldsymbol{z}'} = \sigma_{\boldsymbol{z}}$ in a neighborhood $\mathcal{U}$ of $\boldsymbol{z}$.

As $\mathcal{Z}$ is open in the product manifold, we may shrink $\mathcal{U}$ so that $\mathcal{U} = \mathcal{U}_1 \times \cdots \times \mathcal{U}_L$ with each $\mathcal{U}_j$ path-connected. Fix $i \in [K]$ and let $\tilde{\boldsymbol{z}} \in \mathcal{U}$ satisfy $\tilde{\boldsymbol{z}}_{\sigma(i)} = \boldsymbol{z}_{\sigma(i)}$. Choose a smooth path $\boldsymbol{\gamma} \colon [0, 1] \to \mathcal{U}$ with $\boldsymbol{\gamma}(0) = \boldsymbol{z}$ and $\boldsymbol{\gamma}(1) = \tilde{\boldsymbol{z}}$. By the fundamental theorem of calculus,

$$\boldsymbol{v}_i(\tilde{\boldsymbol{z}}) - \boldsymbol{v}_i(\boldsymbol{z}) = \int_0^1 \frac{\mathrm{d}}{\mathrm{d}t} \boldsymbol{v}_i(\boldsymbol{\gamma}(t)) \, \mathrm{d}t.$$

By the chain rule,

$$\frac{\mathrm{d}}{\mathrm{d}t}\boldsymbol{v}_i(\boldsymbol{\gamma}(t)) = D(\boldsymbol{\pi}_i \circ \boldsymbol{v})_{\boldsymbol{\gamma}(t)} \cdot \dot{\boldsymbol{\gamma}}(t)$$

$$= D_{\sigma(i)}(\boldsymbol{\pi}_i \circ \boldsymbol{v})_{\boldsymbol{\gamma}(t)} \cdot \dot{\boldsymbol{\gamma}}_{\sigma(i)}(t) + \sum_{j \neq \sigma(i)} D_j(\boldsymbol{\pi}_i \circ \boldsymbol{v})_{\boldsymbol{\gamma}(t)} \cdot \dot{\boldsymbol{\gamma}}_j(t).$$

The first term vanishes because $\boldsymbol{\gamma}_{\sigma(i)}(t)$ is constant, and the second vanishes by the structural assumption on $D\boldsymbol{v}$. Thus the integral is zero, and we conclude $\boldsymbol{v}_i(\tilde{\boldsymbol{z}}) = \boldsymbol{v}_i(\boldsymbol{z})$. Hence $\boldsymbol{v}_i$ depends only on the coordinate $\boldsymbol{z}_{\sigma(i)}$, completing the proof. $\qquad\square$

**Theorem 2** (Local identifiability of Type D). *Let $\boldsymbol{g}\colon \mathcal{S} \to \mathcal{X}$ and $\hat{\boldsymbol{g}}\colon \mathcal{Z} \to \mathcal{X}$ be local diffeomorphisms with $\boldsymbol{g}(\mathcal{S}) \subseteq \hat{\boldsymbol{g}}(\mathcal{Z})$. Then $\hat{\boldsymbol{g}}$ is locally disentangled w.r.t. $\boldsymbol{g}$ if:*

*(1) $\mathcal{S} \subseteq \prod_{i=1}^{K} \mathcal{S}_i$ is open, and all factors are Type D independent and irreducible.*

*(2) $\mathcal{Z} \subseteq \prod_{i=1}^{L} \mathcal{Z}_i$ is open with $L = K$, and the factors are independent of Type D.*

*If $L < K$, then $\hat{\boldsymbol{g}}$ is locally partially disentangled.*

*Proof.* Fix an arbitrary point $\boldsymbol{s}^* \in \mathcal{S}$. By the range assumption $\boldsymbol{g}(\mathcal{S}) \subseteq \hat{\boldsymbol{g}}(\mathcal{Z})$, there exists at least one $\boldsymbol{z}^* \in \mathcal{Z}$ such that

$$\boldsymbol{g}(\boldsymbol{s}^*) = \hat{\boldsymbol{g}}(\boldsymbol{z}^*).$$

Since both $\boldsymbol{g}$ and $\hat{\boldsymbol{g}}$ are assumed to be local diffeomorphisms, there exists a neighborhood $\mathcal{U} \subseteq \mathcal{Z}$ of $\boldsymbol{z}^*$ such that, for all $\boldsymbol{z} \in \mathcal{U}$,

$$\hat{\boldsymbol{g}}(\boldsymbol{z}) = \boldsymbol{g} \circ \boldsymbol{v}(\boldsymbol{z}), \tag{13}$$

where we define

$$\boldsymbol{v} \coloneqq \boldsymbol{g}^{-1} \circ \hat{\boldsymbol{g}}\big|_{\mathcal{U}} \colon \mathcal{U} \to (\boldsymbol{g}^{-1} \circ \hat{\boldsymbol{g}})(\mathcal{U}),$$

and $\boldsymbol{g}^{-1}$ denotes the local inverse satisfying $\boldsymbol{v}(\boldsymbol{z}^*) = \boldsymbol{s}^*$. Differentiating gives

$$D\hat{\boldsymbol{g}}_{\boldsymbol{z}} = D\boldsymbol{g}_{\boldsymbol{v}(\boldsymbol{z})} \circ D\boldsymbol{v}_{\boldsymbol{z}}. \tag{14}$$

To obtain matrix representations, choose product-aligned bases on $T_{\boldsymbol{v}(\boldsymbol{z})}(\prod_i \mathcal{S}_i)$ and $T_{\boldsymbol{z}}(\prod_j \mathcal{Z}_j)$, and identify $T_{\hat{\boldsymbol{g}}(\boldsymbol{z})}\mathcal{X}$ and $T_{\boldsymbol{g}(\boldsymbol{v}(\boldsymbol{z}))}\mathcal{X}$ with their natural inclusions into $\mathbb{R}^{d_x}$.

By Type D independence for $\boldsymbol{g}$, the row supports of the partial derivatives $D_i\boldsymbol{g_s}$ and $D_j\boldsymbol{g_s}$ are disjoint whenever $i \neq j$. Thus there is a partition of observation coordinates $[d_x] = \mathcal{R}_1 \cup \cdots \cup \mathcal{R}_K$ such that rows in $\mathcal{R}_i$ depend only on $T_{\boldsymbol{s}_i}\mathcal{S}_i$. Permuting rows by $\boldsymbol{P}$ to group $\mathcal{R}_1, \ldots, \mathcal{R}_K$ consecutively makes $\boldsymbol{A} = \boldsymbol{P}\,D\boldsymbol{g}_{\boldsymbol{v}(\boldsymbol{z})}$ block-row diagonal. Set

$$\boldsymbol{A} \coloneqq \boldsymbol{P}\,D\boldsymbol{g}_{\boldsymbol{v}(\boldsymbol{z})}, \qquad \boldsymbol{B} \coloneqq D\boldsymbol{v}_{\boldsymbol{z}}, \qquad \boldsymbol{C} \coloneqq \boldsymbol{P}\,D\hat{\boldsymbol{g}}_{\boldsymbol{z}},$$

so that $\boldsymbol{C} = \boldsymbol{A}\,\boldsymbol{B}$.

For $k \in [L]$, let $\boldsymbol{B}_{:,k}$ denote the block-columns of $\boldsymbol{B}$ corresponding to $T_{\boldsymbol{z}_k}\mathcal{Z}_k$, and let $\boldsymbol{B}_{:,-k}$ denote the block-columns corresponding to $\bigoplus_{j \neq k} T_{\boldsymbol{z}_j}\mathcal{Z}_j$. Define $\boldsymbol{C}_{:,k}$ and $\boldsymbol{C}_{:,-k}$ analogously as the corresponding block-columns of $\boldsymbol{C}$. Then

$$[\boldsymbol{C}_{:,k} \quad \boldsymbol{C}_{:,-k}] = \begin{bmatrix} \boldsymbol{A}_{1,1} & \boldsymbol{0} & \cdots & \boldsymbol{0} \\ \boldsymbol{0} & \boldsymbol{A}_{2,2} & \cdots & \boldsymbol{0} \\ \vdots & \vdots & \ddots & \vdots \\ \boldsymbol{0} & \boldsymbol{0} & \cdots & \boldsymbol{A}_{K,K} \end{bmatrix} \begin{bmatrix} \boldsymbol{B}_{1,k} & \boldsymbol{B}_{1,-k} \\ \boldsymbol{B}_{2,k} & \boldsymbol{B}_{2,-k} \\ \vdots & \vdots \\ \boldsymbol{B}_{K,k} & \boldsymbol{B}_{K,-k} \end{bmatrix}. \tag{15}$$

By Type D independence for $\hat{\boldsymbol{g}}$, the column supports of $\boldsymbol{C}$ from different target slots are disjoint in observation coordinates, which is preserved by left-multiplication with $\boldsymbol{P}$. Hence the supports of the columns of $\boldsymbol{C}_{:,k}$ are disjoint from those of $\boldsymbol{C}_{:,-k}$, so all pairwise Hadamard products between them

vanish. Denoting the Kronecker product by $\otimes$ and the row-wise Kronecker product (also known as the face-splitting product) by $\odot$, we obtain

$$
\begin{aligned}
\mathbf{0} &= \boldsymbol{C}_{:,k} \odot \boldsymbol{C}_{:,-k} \\
&= (\boldsymbol{A}\boldsymbol{B}_{:,k}) \odot (\boldsymbol{A}\boldsymbol{B}_{:,-k}) \\
&= (\boldsymbol{A} \odot \boldsymbol{A})(\boldsymbol{B}_{:,k} \otimes \boldsymbol{B}_{:,-k}) \\
&= [\boldsymbol{A}_{:,1} \odot \boldsymbol{A}_{:,1} \quad \boldsymbol{A}_{:,2} \odot \boldsymbol{A}_{:,2} \quad \cdots \quad \boldsymbol{A}_{:,K} \odot \boldsymbol{A}_{:,K}]
\begin{bmatrix}
\boldsymbol{B}_{1,k} \otimes \boldsymbol{B}_{1,-k} \\
\boldsymbol{B}_{2,k} \otimes \boldsymbol{B}_{2,-k} \\
\vdots \\
\boldsymbol{B}_{K,k} \otimes \boldsymbol{B}_{K,-k}
\end{bmatrix} \\
&= 
\begin{bmatrix}
(\boldsymbol{A}_{1,1} \odot \boldsymbol{A}_{1,1})(\boldsymbol{B}_{1,k} \otimes \boldsymbol{B}_{1,-k}) \\
(\boldsymbol{A}_{2,2} \odot \boldsymbol{A}_{2,2})(\boldsymbol{B}_{2,k} \otimes \boldsymbol{B}_{2,-k}) \\
\vdots \\
(\boldsymbol{A}_{K,K} \odot \boldsymbol{A}_{K,K})(\boldsymbol{B}_{K,k} \otimes \boldsymbol{B}_{K,-k})
\end{bmatrix}.
\end{aligned}
$$

Here, the third equality uses the mixed-product property, the fourth expands and reorders terms, and the last exploits the block-diagonal structure of $\boldsymbol{A}$. Reversing the mixed-product property yields, for all $i \in [K]$ and $k \in [L]$,

$$
(\boldsymbol{A}_{i,i}\boldsymbol{B}_{i,k}) \odot (\boldsymbol{A}_{i,i}\boldsymbol{B}_{i,-k}) = \mathbf{0}. \tag{16}
$$

Suppose, for a contradiction, that both $\boldsymbol{B}_{i,k}$ and $\boldsymbol{B}_{i,-k}$ are nonzero. Since $\boldsymbol{v}$ is a composition of diffeomorphisms, $\boldsymbol{B}$ is invertible and each $\boldsymbol{B}_{i,:}$ has full row rank. Let us consider two cases (note that $\dim(\mathcal{S}_i) = 0$ and $\dim(\mathcal{Z}_i) = 0$ were categorically excluded in advance):

*Case 1 ($\dim(\mathcal{S}_i) = 1$).* Here $\boldsymbol{B}_{i,:}$ consists of a single row. Choose nonzero scalars $a \in \boldsymbol{B}_{i,k}$ and $b \in \boldsymbol{B}_{i,-k}$. From Equation 16,

$$
(\boldsymbol{A}_{i,i}a) \odot (\boldsymbol{A}_{i,i}b) = \mathbf{0},
$$

which implies $\boldsymbol{A}_{i,i} = \mathbf{0}$, contradicting the assumption that $\boldsymbol{g}$ is a local diffeomorphism.

*Case 2 ($\dim(\mathcal{S}_i) > 1$).* In this case, select columns from $\boldsymbol{B}_{i,k}$ and $\boldsymbol{B}_{i,-k}$ that together form an invertible square matrix $\widetilde{\boldsymbol{B}} = (\widetilde{\boldsymbol{B}}_l, \widetilde{\boldsymbol{B}}_r)$, with $\widetilde{\boldsymbol{B}}_l$ consisting of columns of $\boldsymbol{B}_{i,k}$ and $\widetilde{\boldsymbol{B}}_r$ of $\boldsymbol{B}_{i,-k}$. Then Equation 16 gives

$$
(\boldsymbol{A}_{i,i}\widetilde{\boldsymbol{B}}_l) \odot (\boldsymbol{A}_{i,i}\widetilde{\boldsymbol{B}}_r) = \mathbf{0}.
$$

This implies that $\mathcal{S}_i$ is reducible, since there exists a basis in which $T_{\boldsymbol{s}_i}\mathcal{S}_i$ decomposes into subspaces where all pairwise directional derivatives vanish in the Hadamard product. Hence, either $\boldsymbol{B}_{i,k}$ or $\boldsymbol{B}_{i,-k}$ must be zero.

Repeating the argument for all $i \in [K]$ and $k \in [L]$ shows that each block-row of $\boldsymbol{B}$ contains at most one nonzero block. Since $\boldsymbol{B}$ is invertible, each block-row must contain exactly one nonzero block. Hence, there exists a surjection $\sigma \colon [K] \to [L]$ such that

$$
\boldsymbol{B}_{i,\sigma(i)} \neq \mathbf{0} \quad \text{and} \quad \boldsymbol{B}_{i,j} = \mathbf{0} \ \text{ for } j \neq \sigma(i).
$$

By Lemma 2, it follows that on $\mathcal{U}$, the component $\boldsymbol{v}_i(\boldsymbol{z})$ depends only on $\boldsymbol{z}_{\sigma(i)}$ for every $i \in [K]$. Equivalently, there exist functions

$$
\tilde{\boldsymbol{v}}_i \colon \mathcal{Z}_{\sigma(i)} \to \mathcal{S}_i
$$

such that locally

$$
\boldsymbol{g}^{-1} \circ \hat{\boldsymbol{g}}(\boldsymbol{z}) = \big(\tilde{\boldsymbol{v}}_1(\boldsymbol{z}_{\sigma(1)}), \ldots, \tilde{\boldsymbol{v}}_K(\boldsymbol{z}_{\sigma(K)})\big).
$$

Since $\boldsymbol{s}^*$ was arbitrary and the constructions hold for any $\boldsymbol{z}^*$ satisfying $\boldsymbol{g}(\boldsymbol{s}^*) = \hat{\boldsymbol{g}}(\boldsymbol{z}^*)$, Lemma 1 implies that $\hat{\boldsymbol{g}}$ is locally (partially) disentangled with respect to $\boldsymbol{g}$. $\qquad\square$

## A.4 PROOF OF THEOREM 3

Denote with $\Omega_i(\boldsymbol{s}) \subseteq [d_x]$ the support of the $i$-th column of $\boldsymbol{J_g}(\boldsymbol{s}) = D\boldsymbol{g_s} \colon \mathbb{R}^{d_s} \to \mathbb{R}^{d_x}$ in the standard basis (i.e., $\Omega_i(\boldsymbol{s}) \coloneqq \mathrm{supp}(\boldsymbol{J_g}(\boldsymbol{s})_{:,i})$). Similarly, we use $\widehat{\Omega}_j(\boldsymbol{z})$ for $\boldsymbol{J_{\hat{g}}}(\boldsymbol{z})$. Let $\mathcal{C}_i$ denote the column index set of the $i$-th source factor.

For sets $A, B \subseteq [m]$, write $A \pitchfork B$ iff $A \not\subseteq B$ and $A \not\supseteq B$ (mutual non-inclusion).

**Definition 15** (Mechanistic independence of Type M). *We say that $\mathcal{S}_i$ and $\mathcal{S}_j$ are mechanistically independent of Type M if, for every $\boldsymbol{s} \in \mathcal{S}$,*

$$\forall a \in \mathcal{C}_i, \forall b \in \mathcal{C}_j : \quad \Omega_a(\boldsymbol{s}) \pitchfork \Omega_b(\boldsymbol{s}).$$

**Definition 16** (Reducibility of Type M). *We say that the component $\mathcal{S}_i$ is reducible of Type M if there exist a point $\boldsymbol{s} \in \mathcal{S}$ and a partition $\mathcal{C}_i = \mathcal{A} \cup \mathcal{B}$ such that*

$$\forall a \in \mathcal{A}, \forall b \in \mathcal{B} : \quad \Omega_a(\boldsymbol{s}) \pitchfork \Omega_b(\boldsymbol{s}).$$

**Lemma 3.** *Let $\boldsymbol{C} = \boldsymbol{A}\boldsymbol{B}$, where $\boldsymbol{A} \in \mathbb{R}^{m \times n}$, $\boldsymbol{B} \in \mathbb{R}^{n \times n}$, and $\boldsymbol{C} \in \mathbb{R}^{m \times n}$ are all of full column rank. Define $\mathcal{G}^S(\boldsymbol{A}) := ([n], \mathcal{E}^S)$ with $\mathcal{E}^S = \{(i,j) \in [n]^2 \mid \operatorname{supp}(\boldsymbol{A}_{:,i}) \not\pitchfork \operatorname{supp}(\boldsymbol{A}_{:,j})\}$. If $\|\boldsymbol{C}\|_0 \le \|\boldsymbol{A}\|_0$ and for all $k \in [n]$*

$$\operatorname{supp}(\boldsymbol{C}_{:,k}) \supseteq \bigcup_{i \in \operatorname{supp}(\boldsymbol{B}_{:,k})} \operatorname{supp}(\boldsymbol{A}_{:,i}), \tag{17}$$

*then $\|\boldsymbol{C}\|_0 = \|\boldsymbol{A}\|_0$ and $\mathcal{G}^S(\boldsymbol{C})$ is isomorphic to $\mathcal{G}^S(\boldsymbol{A})$.*

*Proof.* Write $\mathcal{Q}_i := \operatorname{supp}(\boldsymbol{A}_{:,i})$, $\mathcal{R}_k := \operatorname{supp}(\boldsymbol{B}_{:,k})$, and $\mathcal{U}_k := \operatorname{supp}(\boldsymbol{C}_{:,k})$. Since $\boldsymbol{C}_{:,k} = \sum_{i \in \mathcal{R}_k} \boldsymbol{A}_{:,i} B_{i,k}$, we have $\mathcal{U}_k \subseteq \bigcup_{i \in \mathcal{R}_k} \mathcal{Q}_i$, while Equation 17 gives the reverse inclusion; hence

$$\mathcal{U}_k = \bigcup_{i \in \mathcal{R}_k} \mathcal{Q}_i \qquad \forall k \in [n].$$

Because $\boldsymbol{B}$ is invertible, the Leibniz formula for $\det(\boldsymbol{B}) \ne 0$ yields a permutation $\sigma : [n] \to [n]$ with $B_{i,\sigma(i)} \ne 0$ for all $i$, i.e., $i \in \mathcal{R}_{\sigma(i)}$. Thus

$$\mathcal{Q}_i \subseteq \mathcal{U}_{\sigma(i)} \qquad \forall i \in [n].$$

Summing sizes and using $\|\boldsymbol{C}\|_0 \le \|\boldsymbol{A}\|_0$,

$$\sum_i |\mathcal{Q}_i| \le \sum_i |\mathcal{U}_{\sigma(i)}| = \sum_k |\mathcal{U}_k| = \|\boldsymbol{C}\|_0 \le \|\boldsymbol{A}\|_0 = \sum_i |\mathcal{Q}_i|,$$

so equality holds throughout, which forces $|\mathcal{U}_{\sigma(i)}| = |\mathcal{Q}_i|$ and hence $\mathcal{U}_{\sigma(i)} = \mathcal{Q}_i$ for all $i$. This says the column supports of $\boldsymbol{C}$ are exactly those of $\boldsymbol{A}$ up to a relabelling of indices. Since the edge relation in $\mathcal{G}^S(\cdot)$ depends only on mutual non-inclusion of these supports, the bijection $i \mapsto \sigma(i)$ preserves adjacency:

$$\mathcal{Q}_i \pitchfork \mathcal{Q}_j \iff \mathcal{U}_{\sigma(i)} \pitchfork \mathcal{U}_{\sigma(j)}.$$

Hence $\mathcal{G}^S(\boldsymbol{C}) \cong \mathcal{G}^S(\boldsymbol{A})$. $\qquad \square$

**Theorem 3** (Local identifiability of Type M). *Let $\boldsymbol{g} : \mathcal{S} \to \mathcal{X}$ and $\hat{\boldsymbol{g}} : \mathcal{Z} \to \mathcal{X}$ be local diffeomorphisms with $\boldsymbol{g}(\mathcal{S}) \subseteq \hat{\boldsymbol{g}}(\mathcal{Z})$. Then $\hat{\boldsymbol{g}}$ is locally disentangled w.r.t. $\boldsymbol{g}$ if:*

*(1) $\mathcal{S} \subseteq \mathbb{R}^{d_s}$ is open, and the factors are Type M independent and irreducible.*

*(2) $\mathcal{Z} \subseteq \mathbb{R}^{d_s}$ is open, and the factors are independent of Type M.*

*(3) For all $\boldsymbol{s} \in \mathcal{S}$ and $\boldsymbol{z} \in \mathcal{Z}$ with $\boldsymbol{g}(\boldsymbol{s}) = \hat{\boldsymbol{g}}(\boldsymbol{z})$,*

$$\|\boldsymbol{J}_{\hat{\boldsymbol{g}}}(\boldsymbol{z})\|_0 \le \|\boldsymbol{J}_{\boldsymbol{g}}(\boldsymbol{s})\|_0. \tag{5}$$

*(4) For all such pairs,*

$$\widehat{\Omega}_k(\boldsymbol{z}) = \bigcup_{i \in \operatorname{supp}(\boldsymbol{B}_{:,k})} \Omega_i(\boldsymbol{s}), \tag{6}$$

*where $\boldsymbol{B} := \boldsymbol{J}_{\boldsymbol{g}^{-1} \circ \hat{\boldsymbol{g}}}(\boldsymbol{z})$ and $\widehat{\Omega}_k$ mirrors $\Omega_i$ for $\hat{\boldsymbol{g}}$.*

*Proof.* As before, we begin with the identity

$$\hat{\boldsymbol{g}}(\boldsymbol{z}) = \boldsymbol{g} \circ \boldsymbol{v}(\boldsymbol{z}),$$

defined on a neighborhood $\mathcal{U} \subseteq \mathcal{Z}$, where

$$\boldsymbol{v} := \boldsymbol{g}^{-1} \circ \hat{\boldsymbol{g}}|_{\mathcal{U}} : \mathcal{U} \to (\boldsymbol{g}^{-1} \circ \hat{\boldsymbol{g}})(\mathcal{U})$$

is a diffeomorphism that maps a unique $\boldsymbol{z}^* \in \mathcal{U}$ to some initially chosen arbitrary point $\boldsymbol{s}^* \in \mathcal{S}$. Thus, after differentiation we get

$$\boldsymbol{J}_{\hat{\boldsymbol{g}}}(\boldsymbol{z}) = \boldsymbol{J}_{\boldsymbol{g}}(\boldsymbol{v}(\boldsymbol{z}))\boldsymbol{J}_{\boldsymbol{v}}(\boldsymbol{z}),$$

which we write as $\boldsymbol{C} = \boldsymbol{A}\boldsymbol{B}$. Since both $\boldsymbol{g}$ and $\hat{\boldsymbol{g}}$ are local diffeomorphisms into the same observation manifold, $\boldsymbol{B}$ is square and invertible, and $\boldsymbol{A}$, $\boldsymbol{C}$ have full column rank.

Let $\mathcal{R}_i \subseteq [d_s]$ be the column-index set in the $i$-th source block, and define $\mathcal{C}_j \subseteq [d_s]$ analogously for the target blocks. Then $\{\mathcal{R}_i\}_{i=1}^K$ partitions the columns of $\boldsymbol{A}$ and $\{\mathcal{C}_i\}_{i=1}^L$ partitions the columns of $\boldsymbol{C}$.

**Step 1.** *Each column of $\boldsymbol{B}$ has support contained in a single source block.*

Suppose not: then for some column index $k$, the support $\mathrm{supp}(\boldsymbol{B}_{:,k})$ intersects distinct blocks $\mathcal{R}_p \neq \mathcal{R}_q$. By independence of the $\mathcal{S}_i$, $\boldsymbol{B}$ would mix mutually non-inclusive column supports of $\boldsymbol{A}$. Thus, Equation 6 would force a strict increase in the support, which contradicts the assumption that $\|\boldsymbol{C}\|_0 \leq \|\boldsymbol{A}\|_0$. Hence $\mathrm{supp}(\boldsymbol{B}_{:,k}) \subseteq \mathcal{R}_i$ for some $i$. Define $\mathcal{Q}_i := \{q : \mathrm{supp}(\boldsymbol{B}_{:,q}) \subseteq \mathcal{R}_i\}$, i.e., the column set of $\boldsymbol{B}$ supported in $\mathcal{R}_i$.

**Step 2.** *For each $i$, the columns of $\boldsymbol{B}$ supported in $\mathcal{R}_i$ land in a single target block.*

Assume otherwise: then $\mathcal{Q}_i$ meets two distinct $\boldsymbol{C}$-blocks $\mathcal{C}_\alpha$ and $\mathcal{C}_\beta$. Pick $q_\alpha \in \mathcal{C}_\alpha \cap \mathcal{Q}_i$ and $q_\beta \in \mathcal{C}_\beta \cap \mathcal{Q}_i$. By Lemma 3, there are $u_\alpha$ and $u_\beta$ such that $\mathrm{supp}(\boldsymbol{C}_{:,q_\alpha}) = \mathrm{supp}(\boldsymbol{A}_{:,u_\alpha})$ and $\mathrm{supp}(\boldsymbol{C}_{:,q_\beta}) = \mathrm{supp}(\boldsymbol{A}_{:,u_\beta})$. By Equation 6, for every $k \in \mathrm{supp}(\boldsymbol{B}_{:,q_\alpha}) \subseteq \mathcal{R}_i$,

$$\mathrm{supp}(\boldsymbol{A}_{:,u_\alpha}) = \mathrm{supp}(\boldsymbol{C}_{:,q_\alpha}) = \bigcup_{j \in \mathrm{supp}(\boldsymbol{B}_{:,q_\alpha})} \mathrm{supp}(\boldsymbol{A}_{:,j}) \supseteq \mathrm{supp}(\boldsymbol{A}_{:,k}). \tag{18}$$

This implies $u_\alpha \in \mathcal{R}_i$ due to independence of the source factors. If $u_\alpha$ were not in $\mathcal{R}_i$, then $\mathrm{supp}(\boldsymbol{A}_{:,u_\alpha})$ would contain a column support from a different block by Equation 18. Analogously, we get $u_\beta \in \mathcal{R}_i$.

If $u_\alpha = u_\beta$, then $\mathrm{supp}(\boldsymbol{C}_{:,q_\alpha}) = \mathrm{supp}(\boldsymbol{C}_{:,q_\beta})$, contradicting independence of the target blocks. Thus $u_\alpha \neq u_\beta$.

Define $\mathcal{G}_i^S$ with vertex set $\mathcal{R}_i$ and edge set $\mathcal{E} := \{(a,b) \in \mathcal{R}_i \times \mathcal{R}_i \mid \mathrm{supp}(A_{:,a}) \pitchfork \mathrm{supp}(A_{:,b})\}$. By irreducibility of $\mathcal{S}_i$, $\mathcal{G}_i^S$ is connected. Thus, there is a path $u_\alpha = v_0, v_1, \ldots, v_r = u_\beta$ with each consecutive pair comparable (i.e., either $\mathrm{supp}(A_{:,v_i}) \subseteq \mathrm{supp}(A_{:,v_{i+1}})$ or $\mathrm{supp}(A_{:,v_i}) \supseteq \mathrm{supp}(A_{:,v_{i+1}})$). Let $p$ be the first index where the image of $v_p$ (in $\boldsymbol{C}$) leaves $\mathcal{C}_\alpha$. Then $v_{p-1}$ and $v_p$ are comparable but land in different $\boldsymbol{C}$-blocks, giving a containment across $\boldsymbol{C}$-blocks. This contradicts independence of the target factors. Therefore, for each $i$, all columns of $\boldsymbol{B}$ supported in $\mathcal{R}_i$ belong to a single target block. Since $\boldsymbol{B}$ is invertible, repeating the argument for all $i \in [K]$ shows that each block-row of $\boldsymbol{B}$ contains exactly one nonzero block.

Finally, Lemmas 2 and 1 (as in the proof of Theorem 2) imply that $\hat{\boldsymbol{g}}$ is locally disentangled with respect to $\boldsymbol{g}$.

$\square$

**Proposition 3.** *Let $\boldsymbol{A} \in \mathbb{R}^{m \times n}$. For $k \in [n]$, write $\mathcal{R}_k := \mathrm{supp}(\boldsymbol{A}_{:,k}) \subseteq [m]$ and for $i \in [m]$, write $\mathcal{C}_i := \mathrm{supp}(\boldsymbol{A}_{i,:}) \subseteq [n]$. The following are equivalent:*

*(1) (Mutual non-inclusiveness) For all $k \neq \ell$, $\mathcal{R}_k \pitchfork \mathcal{R}_\ell$ (or equivalently, neither $\mathcal{R}_k \subseteq \mathcal{R}_\ell$ nor $\mathcal{R}_\ell \subseteq \mathcal{R}_k$).*

*(2) For every $k \in [n]$,*

$$\{k\} = \bigcap_{i \in \mathcal{R}_k} \mathcal{C}_i.$$

*Proof.* Fix $k \in [n]$. Observe the identity

$$\{\, j \in [n] : \mathcal{R}_k \subseteq \mathcal{R}_j \,\} = \{\, j \in [n] : j \in \mathcal{C}_i \; \forall i \in \mathcal{R}_k \,\}$$
$$= \{\, j \in [n] : A_{ij} \neq 0 \; \forall i \in \mathcal{R}_k \,\}$$
$$= \bigcap_{i \in \mathcal{R}_k} \mathcal{C}_i.$$

Thus (2) is equivalent to $\{k\} = \{\, j : \mathcal{R}_k \subseteq \mathcal{R}_j \,\}$. That is, the only column whose support contains $\mathcal{R}_k$ is $k$ itself. This rules out $\mathcal{R}_k \subseteq \mathcal{R}_j$ for any $j \neq k$, and by symmetry across pairs $(k, \ell)$ yields (1).

Conversely, if (1) holds, then for each $k$ there is no $j \neq k$ with $\mathcal{R}_k \subseteq \mathcal{R}_j$. So by the above identity we get $\bigcap_{i \in \mathcal{R}_k} \mathcal{C}_i = \{k\}$, which is (2). $\qquad\square$

**Remark 4.** *Under the usual convention that $\bigcap_{i \in \varnothing} \mathcal{C}_i = [n]$, both conditions in Proposition 3 forbid zero columns (unless $n = 1$, in which case both are true regardless if the column contains nonzero elements or not).*

**Proposition 4.** *Type M identifiability generalizes Theorem 3.1 from Zheng & Zhang (2023).*

*Proof.* We will show that the assumptions of Theorem 3.1 in Zheng & Zhang (2023) imply the assumptions of Theorem 3 when we pick $\mathcal{S}_i = \mathbb{R}$.

Zheng & Zhang (2023) show that condition *(i)* in Theorem 3.1 implies Equation 14 in their appendix ($\forall (i, j) \in \mathcal{F}, \{i\} \times \mathcal{T}_{j,:} \subset \hat{\mathcal{F}}$), which can be reformulated as Equation 6. Furthermore, Proposition 3 establishes that *structural sparsity* (condition *(ii)* in Theorem 3.1) is equivalent to mutual non-inclusion. Thus, structural sparsity implies Type M independence of the source factors. The sparsity gap (Equation 5) is not explicitly listed in Theorem 3.1 but required throughout their entire work. Finally, for one-dimensional factors, Type M irreducibility is vacuously true, and by Lemma 3 Type M independence of the target factors holds automatically. $\qquad\square$

## A.5 PROOF OF THEOREM 4

For $s \in \mathcal{S}$, denote by $\rho_{\mathfrak{B}}^+(s)$ the minimal $\ell_0$-norm (i.e. the number of nonzero entries) of the matrix representing $D\boldsymbol{g}_{\boldsymbol{s}} : T_{\boldsymbol{s}} \mathcal{S} \to T_{\boldsymbol{g}(\boldsymbol{s})} \mathcal{X}$ when expressed in a basis of $T_{\boldsymbol{s}} \mathcal{S}$ that is *aligned* with the decomposition $\mathfrak{B}$ and in the canonical basis of $T_{\boldsymbol{g}(\boldsymbol{s})} \mathcal{X}$ induced by its embedding in $\mathbb{R}^{d_x}$. Conversely, define $\rho_{\mathfrak{B}}^-(s)$ as the infimum of the $\ell_0$-norm of $D\boldsymbol{g}_{\boldsymbol{s}}$ taken over all choices of basis of $T_{\boldsymbol{s}} \mathcal{S}$ that do *not* respect the decomposition $\mathfrak{B}$. Analogously, we define $\rho_{\mathfrak{B}_i}^+(s)$ and $\rho_{\mathfrak{B}_i}^-(s)$ based on $D_i \boldsymbol{g}_{\boldsymbol{s}}$, where $\mathfrak{B}_i$ is a decomposition of $T_{\boldsymbol{s}_i} \mathcal{S}_i$.

**Definition 17** (Mechanistic independence of Type S). *We say that the subspaces $\mathcal{S}_i$ are mechanistically independent of Type S if, for every $\boldsymbol{s} \in \mathcal{S}$,*

$$\rho_{\mathfrak{B}}^+(\boldsymbol{s}) \; < \; \rho_{\mathfrak{B}}^-(\boldsymbol{s}), \quad where \quad \mathfrak{B} := \bigoplus_{i \in [K]} T_{\boldsymbol{s}_i} \mathcal{S}_i.$$

**Definition 18** (Reducibility of Type S). *We say that the component $\mathcal{S}_i$ is reducible of Type S if there exist $\boldsymbol{s} \in \mathcal{S}$ and a nontrivial decomposition $T_{\boldsymbol{s}_i} \mathcal{S}_i = U \oplus V =: \mathfrak{B}_i$ such that*

$$\rho_{\mathfrak{B}_i}^+(\boldsymbol{s}) \; < \; \rho_{\mathfrak{B}_i}^-(\boldsymbol{s}).$$

*Otherwise, we call $\mathcal{S}_i$ irreducible of Type S.*

**Theorem 4** (Local identifiability of Type S). *Let $\boldsymbol{g} : \mathcal{S} \to \mathcal{X}$ and $\hat{\boldsymbol{g}} : \mathcal{Z} \to \mathcal{X}$ be local diffeomorphisms with $\boldsymbol{g}(\mathcal{S}) \subseteq \hat{\boldsymbol{g}}(\mathcal{Z})$. Then $\hat{\boldsymbol{g}}$ is locally disentangled w.r.t. $\boldsymbol{g}$ if:*

*(1) $\mathcal{S} \subseteq \prod_{i=1}^{K} \mathcal{S}_i$ is open, and the factors $\mathcal{S}_i$ are Type S independent and irreducible.*

*(2) $\mathcal{Z} \subseteq \prod_{j=1}^{L} \mathcal{Z}_j$ is open with $L = K$, and the factors $\mathcal{Z}_j$ are independent of Type S.*

*If $L < K$, then $\hat{\boldsymbol{g}}$ is locally partially disentangled.*

*Proof.* On a neighborhood $\mathcal{U} \subseteq \mathcal{Z}$ define the diffeomorphism

$$\boldsymbol{v} := \boldsymbol{g}^{-1} \circ \hat{\boldsymbol{g}}\big|_{\mathcal{U}} : \mathcal{U} \to (\boldsymbol{g}^{-1} \circ \hat{\boldsymbol{g}})(\mathcal{U}),$$

so that $\hat{\boldsymbol{g}} = \boldsymbol{g} \circ \boldsymbol{v}$ on $\mathcal{U}$. Hence

$$D\hat{\boldsymbol{g}}_{\boldsymbol{z}} = D\boldsymbol{g}_{\boldsymbol{v}(\boldsymbol{z})} \circ D\boldsymbol{v}_{\boldsymbol{z}}. \tag{19}$$

Fix product-splitting bases for $T_{\boldsymbol{v}(\boldsymbol{z})}(\prod_i \mathcal{S}_i)$ and $T_{\boldsymbol{z}}(\prod_j \mathcal{Z}_j)$ that minimize the $\ell_0$-sparsity of $D\boldsymbol{g}_{\boldsymbol{v}(\boldsymbol{z})}$ and $D\hat{\boldsymbol{g}}_{\boldsymbol{z}}$ respectively. In these bases, write Equation 19 as $\boldsymbol{C} = \boldsymbol{A}\boldsymbol{B}$. Since both $\boldsymbol{g}$ and $\hat{\boldsymbol{g}}$ are local diffeomorphisms into the same observation manifold, $\boldsymbol{B}$ is square and invertible. Let $\mathcal{R}_i \subseteq [d_s]$ be the column-index set spanning $T_{\boldsymbol{s}_i} \mathcal{S}_i$, and define $\mathcal{C}_j \subseteq [d_s]$ analogously for $T_{\boldsymbol{z}_j} \mathcal{Z}_j$.

**Step 1.** *Each column of $\boldsymbol{B}$ has support contained in a single source block.*

Suppose not: then for some column index $k$, the support $\mathrm{supp}(\boldsymbol{B}_{:,k})$ intersects distinct blocks $\mathcal{R}_p \neq \mathcal{R}_q$. By independence of the $\mathcal{S}_i$, any basis change of $D\boldsymbol{g}_{\boldsymbol{s}}$ that mixes coordinates from different source blocks worsens the $\ell_0$-sparsity after multiplication. Equivalently,

$$\|\boldsymbol{A}\|_0 < \|\boldsymbol{A}\boldsymbol{B}\|_0 = \|\boldsymbol{C}\|_0.$$

This contradicts the assumption that the chosen basis for $D\hat{\boldsymbol{g}}_{\boldsymbol{z}}$ is $\ell_0$-minimal, since independence of the $\mathcal{Z}_j$ implies that the lowest $\ell_0$-norm is achieved in a product-splitting basis (up to reordering of the basis vectors). Hence $\mathrm{supp}(\boldsymbol{B}_{:,k}) \subseteq \mathcal{R}_i$ for some $i$.

**Step 2.** *For each $i$, the columns of $\boldsymbol{B}$ supported in $\mathcal{R}_i$ land in a single target block.*

Assume otherwise: then there exists $i \in [K]$ and columns $p \in \mathcal{C}_k$ and $q \in \mathcal{C}_{-k} := \bigcup_{j \neq k} \mathcal{C}_j$ such that both $\boldsymbol{B}_{:,p}$ and $\boldsymbol{B}_{:,q}$ are supported in $\mathcal{R}_i$. Now consider two cases (with $\dim(\mathcal{S}_i) = 0$ and $\dim(\mathcal{Z}_i) = 0$ excluded a priori):

*Case 1 ($\dim(\mathcal{S}_i) = 1$).* Then $\mathcal{R}_i = \{r\}$ and for nonzero scalars $B_{r,p}, B_{r,q}$ we have

$$\mathrm{supp}(\boldsymbol{C}_{:,p}) = \mathrm{supp}(\boldsymbol{A}_{:,r} B_{r,p}) = \mathrm{supp}(\boldsymbol{A}_{:,r} B_{r,q}) = \mathrm{supp}(\boldsymbol{C}_{:,q}).$$

However, a necessary requirement for independence of the target factors is

$$\mathrm{supp}(\boldsymbol{C}_{:,p}) \pitchfork \mathrm{supp}(\boldsymbol{C}_{:,q}),$$

since otherwise a cross-block mixing can be constructed involving $\boldsymbol{C}_{:,p}$ and $\boldsymbol{C}_{:,q}$ which leaves the overall support unchanged. This contradicts the earlier result that $\mathrm{supp}(\boldsymbol{C}_{:,p}) = \mathrm{supp}(\boldsymbol{C}_{:,q})$.

*Case 2 ($\dim(\mathcal{S}_i) > 1$).* The full row rank of $\boldsymbol{B}_{\mathcal{R}_i,:}$ yields an invertible square submatrix $\widetilde{\boldsymbol{B}}$ formed from columns in $\mathcal{C}_k$ and $\mathcal{C}_{-k}$ such that

$$\boldsymbol{A}_{:,\mathcal{R}_i} \widetilde{\boldsymbol{B}} = [\widetilde{\boldsymbol{A}}_1, \widetilde{\boldsymbol{A}}_2],$$

where $\widetilde{\boldsymbol{A}}_1$ and $\widetilde{\boldsymbol{A}}_2$ are submatrices of $\boldsymbol{C}_{:,\mathcal{C}_k}$ and $\boldsymbol{C}_{:,\mathcal{C}_{-k}}$, respectively. By independence of the $\mathcal{Z}_j$,

$$\hat{\rho}_{\widehat{\mathfrak{B}}}^{+}(\boldsymbol{z}) < \hat{\rho}_{\widehat{\mathfrak{B}}}^{-}(\boldsymbol{z}), \quad \text{where} \quad \widehat{\mathfrak{B}} := \bigoplus_{i \in [L]} T_{\boldsymbol{z}_i} \mathcal{Z}_i.$$

This forces

$$\hat{\rho}_{\widehat{\mathfrak{B}}}^{+}(\boldsymbol{z}) = \|\boldsymbol{C}\|_0 = \|[\widetilde{\boldsymbol{A}}_1, \widetilde{\boldsymbol{A}}_2]\|_0 + c < \hat{\rho}_{\widehat{\mathfrak{B}}}^{-}(\boldsymbol{z}) \leq \inf_{\boldsymbol{G} \notin \{\text{block-respecting}\}} \|[\widetilde{\boldsymbol{A}}_1, \widetilde{\boldsymbol{A}}_2]\boldsymbol{G}\|_0 + c,$$

where $c \geq 0$ is the number of nonzero entries of $\boldsymbol{C}$ outside $[\widetilde{\boldsymbol{A}}_1, \widetilde{\boldsymbol{A}}_2]$. Since $\boldsymbol{C}$ has minimal support, there is no basis transformation reducing the $\ell_0$-norm of $\widetilde{\boldsymbol{A}}_1$ or $\widetilde{\boldsymbol{A}}_2$ individually. Thus

$$\rho_{\mathfrak{B}_i}^{+}(\boldsymbol{v}(\boldsymbol{z})) \leq \|\boldsymbol{A}_{:,\mathcal{R}_i} \widetilde{\boldsymbol{B}}\|_0 < \inf_{\boldsymbol{G} \notin \{\text{block-respecting}\}} \|[\widetilde{\boldsymbol{A}}_1, \widetilde{\boldsymbol{A}}_2]\boldsymbol{G}\|_0 = \rho_{\mathfrak{B}_i}^{-}(\boldsymbol{v}(\boldsymbol{z})),$$

contradicting irreducibility of $\mathcal{S}_i$.

Hence, for each $i$, all columns of $\boldsymbol{B}$ supported in $\mathcal{R}_i$ belong to a single target block. Repeating the argument for all $i \in [K]$ shows that each block-row of $\boldsymbol{B}$ contains exactly one nonzero block (since $\boldsymbol{B}$ is invertible).

Finally, Lemmas 2 and 1 (as in the proof of Theorem 2) imply that $\hat{\boldsymbol{g}}$ is locally (partially) disentangled with respect to $\boldsymbol{g}$. $\qquad\square$

A.6   PROOF OF THEOREM 5

**Definition 19** (Mechanistic independence of Type $H_n$). *Let $\mathcal{S} \subseteq \prod_{i=1}^K \mathcal{S}_i$ be a smooth manifold, and let $\boldsymbol{g} \colon \mathcal{S} \to \mathcal{X}$ be of class $C^n$ with $n \geq 2$. $\mathcal{S}_i$ and $\mathcal{S}_j$ are said to be* mechanistically independent *of Type $H_n$ if, for all $\boldsymbol{s} \in \mathcal{S}$,*

$$D_{i,j}^n \boldsymbol{g_s} = \boldsymbol{0}. \tag{20}$$

**Definition 20** (Reducibility of Type $H_n$). *We say that the component $\mathcal{S}_i$ is reducible of Type $H_n$ if there exists $\boldsymbol{s} \in \mathcal{S}$ such that either $D_{i,i}^n \boldsymbol{g_s} = \boldsymbol{0}$ or there exists a nontrivial splitting $T_{\boldsymbol{s}_i} \mathcal{S}_i = U \oplus V$ such that for all $\boldsymbol{\xi} \in U$, $\boldsymbol{\eta} \in V$, and $\boldsymbol{\zeta}_k \in T_{\boldsymbol{s}} \mathcal{S}$ for $k \in [n-2]$,*

$$D_{i,i}^n \boldsymbol{g_s}(\boldsymbol{\xi}, \boldsymbol{\eta}, \boldsymbol{\zeta}_1, \dots, \boldsymbol{\zeta}_{n-2}) = \boldsymbol{0}. \tag{21}$$

**Definition 21** (Separability of $n$-th order). *We say that $\boldsymbol{g} \colon \mathcal{S} \to \mathcal{X}$ is separable of order $n$ if there exists $\boldsymbol{s} \in \mathcal{S}$ such that, for all $i \in [K]$, the image of $D_{i,i}^n \boldsymbol{g_s}$ intersects trivially with*

$$\mathrm{span}\Big\{ D_{j,j}^n \boldsymbol{g_s}, \ j \neq i; \ D^k \boldsymbol{g_s}, \ 1 \leq k \leq n-1 \Big\}.$$

**Lemma 4.** *Let $V$ be a finite-dimensional vector space with $\dim(V) \geq 2$, and suppose $W_1, \dots, W_n$ with $n \geq 2$ are subspaces of $V$ such that $W_1 + \cdots + W_n = V$. Assume that there exist indices $i \neq j$ that satisfy $W_i \neq \{\boldsymbol{0}\}$ and $W_j \neq \{\boldsymbol{0}\}$. Then there exist nonzero subspaces $U_1$ and $U_2$ of $V$ such that*

$$V = U_1 \oplus U_2,$$

*with $U_1 \subseteq W_i$ and $U_2 \subseteq \sum_{k \neq i} W_k$.*

*Proof.* Set $C := \sum_{k \neq i} W_k$ and $V_0 := W_i \cap C$. Then choose complements

$$W_i = V_0 \oplus V_1 \quad \text{and} \quad C = V_0 \oplus V_2$$

for some subspaces $V_1 \subseteq W_i$ and $V_2 \subseteq C$. Then

$$V = W_i + C = (V_0 \oplus V_1) + (V_0 \oplus V_2) = V_0 \oplus V_1 \oplus V_2,$$

and the sum is direct because $V_1 \cap V_2 = \{\boldsymbol{0}\}$ and $V_0 \cap (V_1 + V_2) = \{\boldsymbol{0}\}$.

We now choose $U_1$ and $U_2$ case by case.

*Case 1: $V_1 \neq \{\boldsymbol{0}\}$ and $V_2 \neq \{\boldsymbol{0}\}$.* Set $U_1 := V_1 \subseteq W_i$ and $U_2 := V_0 \oplus V_2 \subseteq C$. Then $U_1 \oplus U_2 = V_1 \oplus (V_0 \oplus V_2) = V$, and both $U_1, U_2$ are nonzero.

*Case 2: $V_1 \neq \{\boldsymbol{0}\}$ and $V_2 = \{\boldsymbol{0}\}$.* Then $C = V_0$ and, since $W_j \subseteq C$ with $W_j \neq \{\boldsymbol{0}\}$, we have $V_0 \neq \{\boldsymbol{0}\}$. Set $U_1 := V_1 \subseteq W_i$ and $U_2 := V_0 \subseteq C$. Again $U_1 \oplus U_2 = V_1 \oplus V_0 = V$, with both nonzero.

*Case 3: $V_1 = \{\boldsymbol{0}\}$ and $V_2 \neq \{\boldsymbol{0}\}$.* Then $W_i = V_0$, hence $V_0 \neq \{\boldsymbol{0}\}$ because $W_i \neq \{\boldsymbol{0}\}$. Set $U_1 := V_0 \subseteq W_i$ and $U_2 := V_2 \subseteq C$. We have $U_1 \oplus U_2 = V_0 \oplus V_2 = V$, both nonzero.

*Case 4: $V_1 = \{\boldsymbol{0}\}$ and $V_2 = \{\boldsymbol{0}\}$.* Then $W_i = C = V_0$. In particular $W_i = C = V$. Since $\dim(V) \geq 2$, choose a decomposition $V = A \oplus B$ with $A, B \neq \{\boldsymbol{0}\}$. Taking $U_1 := A \subseteq W_i$ and $U_2 := B \subseteq C$ yields the claim.

In all cases we obtain nonzero subspaces $U_1 \subseteq W_i$ and $U_2 \subseteq C = \sum_{k \neq i} W_k$ with $V = U_1 \oplus U_2$, as required. $\square$

**Theorem 5** (Local identifiability of Type $H_n$). *Let $\boldsymbol{g} \colon \mathcal{S} \to \mathcal{X}$ and $\hat{\boldsymbol{g}} \colon \mathcal{Z} \to \mathcal{X}$ be local $C^n$-diffeomorphisms with $n \geq 2$ satisfying $\boldsymbol{g}(\mathcal{S}) \subseteq \hat{\boldsymbol{g}}(\mathcal{Z})$. Then $\hat{\boldsymbol{g}}$ is locally disentangled w.r.t. $\boldsymbol{g}$ if:*

(1) *$\mathcal{S} \subseteq \prod_{i=1}^K \mathcal{S}_i$ is open, and the factors are Type $H_n$ independent and irreducible.*

(2) *$\mathcal{Z} \subseteq \prod_{j=1}^L \mathcal{Z}_j$ is open with $L = K$, and the factors are independent of Type $H_n$.*

(3) *$\boldsymbol{g}$ is separable of order $n$.*

*If $L < K$, then $\hat{g}$ is locally partially disentangled.*

*Proof.* Let $s^* \in \mathcal{S}$ be arbitrary, and choose $z^* \in \mathcal{Z}$ such that
$$g(s^*) = \hat{g}(z^*).$$
Since $g$ and $\hat{g}$ are local diffeomorphisms, there exists a neighborhood $\mathcal{U} \subseteq \mathcal{Z}$ of $z^*$ on which we may write
$$\hat{g} = g \circ v,$$
where
$$v := g^{-1} \circ \hat{g}\big|_{\mathcal{U}} : \mathcal{U} \to (g^{-1} \circ \hat{g})(\mathcal{U}) \quad \text{satisfies} \quad v(z^*) = s^*.$$
Fix $n \geq 2$. For $z \in \mathcal{U}$, the higher-order chain rule gives
$$D^n \hat{g}_z = \sum_{\pi \in \mathcal{P}([n])} D^{|\pi|} g_{v(z)} \big( D^{|B|} v_z \big)_{B \in \pi}, \tag{22}$$
where $\mathcal{P}([n])$ denotes the set of partitions of $\{1, \ldots, n\}$.

On the left-hand side of Equation 22, mechanistic independence of the $\mathcal{Z}_i$ implies that all mixed derivatives of $\hat{g}$ vanish:
$$D_{i,j}^n \hat{g}_z = \mathbf{0}, \qquad i \neq j \in [L].$$
Now restrict Equation 22 to this mixed derivative and consider the right-hand side. Mechanistic independence of the $\mathcal{S}_i$ implies that the highest-order term (corresponding to $\pi = \{1, \ldots, n\}$) can be split up, and all mixed derivatives $D_{k,l}^n g_{v(z)}$ vanish:
$$D^n g_{v(z)} \big( D_i v_z, D_j v_z, \underbrace{D v_z, \ldots, D v_z}_{n-2 \text{ times}} \big) = \sum_{k \in [K]} D_{k,k}^n g_{v(z)} \big( D_i (\pi_k \circ v)_z, D_j (\pi_k \circ v)_z, \underbrace{D v_z, \ldots, D v_z}_{n-2 \text{ times}} \big),$$
where $\pi_k$ denotes the projection onto the $k$-th slot.

By separability (Defn. 21), the image of $D_{k,k}^n g_{v(z)}$ intersects the images of all other derivative terms on the right-hand side of Equation 22 only at zero. Hence they cannot cancel and each individual term in the sum must be zero. Therefore, for each $k \in [K]$, we obtain
$$D_{k,k}^n g_{v(z)} \big( D_i (\pi_k \circ v)_z, D_j (\pi_k \circ v)_z, \underbrace{D v_z, \ldots, D v_z}_{n-2 \text{ times}} \big) = \mathbf{0}. \tag{23}$$

Now assume, for a contradiction, that there exist $\alpha \in T_{z_i} \mathcal{Z}_i$ and $\beta \in T_{z_j} \mathcal{Z}_j$ such that
$$D_i (\pi_k \circ v)_z(\alpha) \neq \mathbf{0} \quad \text{and} \quad D_j (\pi_k \circ v)_z(\beta) \neq \mathbf{0}.$$

We distinguish two cases (recall that $\dim(\mathcal{S}_i) = 0$ and $\dim(\mathcal{Z}_i) = 0$ were excluded by assumption):

*Case 1:* $\dim(\mathcal{S}_k) = 1$. Then Equation 23 implies $D_{k,k}^n g_{v(z)} = \mathbf{0}$, contradicting irreducibility.

*Case 2:* $\dim(\mathcal{S}_k) > 1$. Define
$$W_i := \text{im}\big(D_i (\pi_k \circ v)_z\big).$$
Since $v$ is a composition of local diffeomorphisms, $D(\pi_k \circ v)_z$ is surjective, hence
$$T_{v_k(z)} \mathcal{S}_k = W_1 + \cdots + W_L.$$
By Lemma 4, we can decompose
$$T_{v_k(z)} \mathcal{S}_k = U_1 \oplus U_2$$
with nontrivial tangent subspaces $U_1 \subseteq W_i$ and $U_2 \subseteq \sum_{j \neq i} W_j$. From Equation 23 we then have, for all $\xi \in U_1$ and $\eta \in U_2$,
$$D_{k,k}^n g_{v(z)} \big( \xi, \eta, \zeta_1, \ldots, \zeta_{n-2} \big) = \mathbf{0},$$
where $\zeta_\ell \in T_{v(z)} \mathcal{S}$ are arbitrary. This implies that $\mathcal{S}_k$ is reducible, a contradiction.

Therefore, for each $k \in [K]$ there is at most one $i \in [L]$ such that
$$D_i \big( \pi_k \circ v \big)_z \neq \mathbf{0}.$$
Since $D v_z$ is an isomorphism, at least one such $i$ must exist. Applying Lemmas 2 and 1, as in the proof of Theorem 2, we obtain a surjection $\sigma : [K] \to [L]$ with the disentanglement property.

Hence $\hat{g}$ is locally (partially) disentangled with respect to $g$. $\qquad\square$

A.7   PROOFS OF GRAPH-THEORETICAL RELATIONS

**Proposition 5.** *Let $\boldsymbol{A} \in \mathbb{R}^{m \times n}$ have full column rank and define $\mathcal{G}(\boldsymbol{A}) = ([n], \mathcal{E})$, $\mathcal{E} = \{(i, j) \in [n]^2 \mid \boldsymbol{A}_{:,i} \odot \boldsymbol{A}_{:,j} \neq \boldsymbol{0}\}$. For a fixed integer $K \geq 1$ the following are equivalent:*

*(i) For any invertible $\boldsymbol{B} \in \mathbb{R}^{n \times n}$ the maximal number of connected components of $\mathcal{G}(\boldsymbol{AB})$ is $K$.*

*(ii) There are a permutation matrix $\boldsymbol{P}$ and an invertible matrix $\boldsymbol{B}$ such that*

$$\boldsymbol{PAB} = \mathrm{diag}\big(\boldsymbol{A}^{(1)}, \ldots, \boldsymbol{A}^{(K)}\big),$$

*and no other $\boldsymbol{P}'$, $\boldsymbol{B}'$ such that $\boldsymbol{P}'\boldsymbol{AB}'$ is block-diagonal with $K+1$ blocks on the diagonal.*

*(iii) There exists an invertible $\boldsymbol{B}$ such that $\boldsymbol{AB}$ is compositional with $K$ irreducible mechanisms in the sense of Definitions 1 and 5 of Brady et al. (2023).*

*(iv) There is a partition $[m] = \mathcal{Q}_1 \cup \cdots \cup \mathcal{Q}_K$ with $\mathcal{Q}_k \neq \varnothing$ such that*

$$\mathrm{rank}(\boldsymbol{A}) = \sum_{k=1}^{K} \mathrm{rank}\big(\boldsymbol{A}_{\mathcal{Q}_k,:}\big), \qquad \mathrm{rank}(\boldsymbol{A}_{\mathcal{Q}_k,:}) \geq 1 \;\; \forall k,$$

*and no partition of $[m]$ into $K + 1$ non-empty sets satisfies this equality.*

*Proof.* Throughout, all ranks are column–ranks. For a matrix $\boldsymbol{X}$, let $\mathrm{row}(\boldsymbol{X})$ denote its row space and let $\mathrm{supp}(\boldsymbol{X})$ be the set of row indices whose corresponding rows are non–zero. Multiplication by an invertible matrix or a permutation matrix preserves rank and does not change the edge–relation that defines the graph $\mathcal{G}(\cdot)$.

(i) $\implies$ (ii)

Statement (i) asserts that there exists a $\boldsymbol{B} \in \mathbb{R}^{n \times n}$ such that $\mathcal{G}(\boldsymbol{AB})$ possesses exactly $K$ connected components. Let $\mathcal{C}_1, \ldots, \mathcal{C}_K \subset [n]$ be the vertex sets of these components and put $\mathcal{R}_k := \bigcup_{i \in \mathcal{C}_k} \mathrm{supp}\big((\boldsymbol{AB})_{:,i}\big) \subseteq [m]$. Without loss of generality we can assume that $\mathcal{C}_1, \ldots, \mathcal{C}_K$ appear in contiguous order. Otherwise, permute the columns of $\boldsymbol{B}$ first.

Because different components have disjoint row supports (otherwise there would be a connecting edge), the sets $\mathcal{R}_1, \ldots, \mathcal{R}_K$ are mutually disjoint. Permute the rows so that $\mathcal{R}_1, \ldots, \mathcal{R}_K$ appear contiguously and denote the corresponding permutation matrix by $\boldsymbol{P}$. Then $\boldsymbol{PAB}$ is block–diagonal with exactly $K$ diagonal blocks. Note that any zero rows of $\boldsymbol{AB}$ can be placed arbitrarily.

If, contrary to the minimality clause of (ii), another pair $\boldsymbol{P}'$, $\boldsymbol{B}'$ produced $K + 1$ diagonal blocks, the graph $\mathcal{G}(\boldsymbol{AB}')$ would contain at least $K + 1$ connected components, contradicting (i). Therefore (ii) holds.

(ii) $\implies$ (iii)

Write $\boldsymbol{PAB} = \mathrm{diag}\big(\boldsymbol{A}^{(1)}, \ldots, \boldsymbol{A}^{(K)}\big)$ as in (ii) and set $\boldsymbol{M}^{(k)} := (\boldsymbol{AB})_{\mathcal{R}_k,:}$ $(k = 1, \ldots, K)$ with $\mathcal{R}_k$ as before. The matrices $\boldsymbol{M}^{(k)}$ have pairwise disjoint row supports, so they constitute $K$ *mechanisms* and $\boldsymbol{AB}$ is *compositional*.

Assume that one mechanism, say $\boldsymbol{M}^{(1)}$, were reducible. Then its row support could be partitioned into two non–empty sets whose row spaces are independent, yielding another decomposition of $\boldsymbol{P}'\boldsymbol{AB}'$ into $K + 1$ diagonal blocks. This contradicts the minimality property in (ii). Thus every mechanism is irreducible and (iii) follows.

(iii) $\implies$ (iv)

Since $\boldsymbol{AB}$ has $K$ compositional mechanisms, there are disjoint $\mathcal{R}_1, \ldots, \mathcal{R}_K \subseteq [m]$. Add zero rows of $\boldsymbol{AB}$ arbitrarily to $\mathcal{R}_i$ denoted by $\mathcal{Q}_i$ (i.e., $\mathcal{R}_i \subseteq \mathcal{Q}_i$) such that $\mathcal{Q}_1, \ldots, \mathcal{Q}'_K$ partition $[m]$. Then $\mathrm{rank}(\boldsymbol{AB}) = \sum_{k=1}^{K} \mathrm{rank}\big((\boldsymbol{AB})_{\mathcal{Q}_k,:}\big)$. and $\mathrm{rank}\big((\boldsymbol{AB})_{\mathcal{Q}_k,:}\big) \geq 1$.

Suppose a refinement $[m] = \mathcal{Q}_1' \cup \cdots \cup \mathcal{Q}_{K+1}'$ also satisfied the same rank identity. Then there is a $\boldsymbol{B}' \in \mathbb{R}^{n \times n}$ such that $\boldsymbol{AB}'$ has $K+1$ compositional machanisms. Next, we show by contradiction that if $\boldsymbol{AB}$ has $K$ compositional and irreducible mechanisms then there is no invertible $\boldsymbol{B}' \in \mathbb{R}^{n \times n}$ such that $\boldsymbol{AB}'$ has more than $K$ compositional mechanisms establishing (iv).

Suppose such a $\boldsymbol{B}'$ existed. Denote with $\{\mathcal{R}_j'\}_{j=1}^{K'}$ (with $K' > K$) the row sets that constitute the compositional mechanisms of $\boldsymbol{AB}'$, respectively.

According to the pigeonhole principle there is at least one $\mathcal{R}_i$ which has elements in multiple $\mathcal{R}_j'$. Denote with $\mathcal{U}_{ij} = \mathcal{R}_i \cap \mathcal{R}_j'$. Then $\mathrm{rank}(\boldsymbol{A}_{\mathcal{R}_i,:}) = \mathrm{rank}(\boldsymbol{A}_{\mathcal{R}_i,:}\boldsymbol{B}) = \sum_j \mathrm{rank}(\boldsymbol{A}_{\mathcal{U}_{i,j},:}\boldsymbol{B}) = \sum_j \mathrm{rank}(\boldsymbol{A}_{\mathcal{U}_{i,j},:})$, which contradicts the irreducibility assumption. Thus, there is no basis in which $\boldsymbol{A}$ has more than $K$ compositional mechanisms.

(iv) $\implies$ (i)

Assume (iv) with partition $[m] = \mathcal{Q}_1 \cup \cdots \cup \mathcal{Q}_K$.

Permute rows so that $\mathcal{Q}_1, \ldots, \mathcal{Q}_K$ are consecutive; call the permutation matrix $\boldsymbol{P}$. Because the row spaces $\mathrm{row}(\boldsymbol{A}_{\mathcal{Q}_k,:})$ are pairwise independent, one may choose a column basis aligned with them, yielding $\boldsymbol{B} \in \mathbb{R}^{n \times n}$ with $\boldsymbol{PAB} = \mathrm{diag}(\boldsymbol{A}^{(1)}, \ldots, \boldsymbol{A}^{(K)})$. Consequently $\mathcal{G}(\boldsymbol{PAB}) = \mathcal{G}(\boldsymbol{AB})$ has at least $K$ connected components.

Now, let $\boldsymbol{B}$ be arbitrary and suppose $\mathcal{G}(\boldsymbol{AB})$ had $K+1$ connected components with vertex sets $\mathcal{C}_1', \ldots, \mathcal{C}_{K+1}'$. As before set $\mathcal{R}_k' := \bigcup_{i \in \mathcal{C}_k'} \mathrm{supp}((\boldsymbol{AB})_{:,i}) \subset [m]$. Disjointness of components implies $[m] = \mathcal{R}_1' \cup \cdots \cup \mathcal{R}_{K+1}'$ and, as before,

$$\mathrm{rank}(\boldsymbol{A}) = \sum_{k=1}^{K+1} \mathrm{rank}\left(\boldsymbol{A}_{\mathcal{R}_k',:}\right),$$

contradicting the minimality clause in (i). Therefore every invertible $\boldsymbol{B}$ produces at most $K$ connected components.

We have established the chain of implications

$$\text{(i)} \implies \text{(ii)} \implies \text{(iv)} \implies \text{(iii)} \implies \text{(i)},$$

hence all four statements are equivalent. $\qquad\square$

## B  EXAMPLES

**Example 1** (Type M and Type S mechanistic independence vs. reducibility)**.** *This example illustrates Type M and Type S mechanistic independence and reducibility. We display four Jacobians, each written in a basis aligned with a given product decomposition of the source tangent space. Block columns (corresponding to distinct source components) are separated by vertical rules:*

$$\boldsymbol{A} = \begin{bmatrix} 1 & 0 \\ 1 & 0 \\ -2 & 1 \\ -1 & 1 \\ 1 & 1 \\ 2 & 1 \\ 0 & 1 \\ 0 & 1 \end{bmatrix} \qquad \boldsymbol{B} = \begin{bmatrix} 1 & 0 & -1 \\ 1 & 0 & 0 \\ -1 & 1 & 0 \\ 0 & 1 & 0 \\ 0 & -1 & 1 \\ 0 & 0 & 1 \end{bmatrix}$$

$$\boldsymbol{C} = \begin{bmatrix} 1 & 0 & 1 \\ 1 & 0 & 0 \\ -1 & 1 & 0 \\ 0 & 1 & 0 \\ 0 & -1 & 1 \\ 0 & 0 & 1 \end{bmatrix} \qquad \boldsymbol{D} = \begin{bmatrix} -1 & 1 & 0 & 0 \\ 1 & 0 & 0 & 0 \\ 1 & 2 & 0 & 0 \\ 0 & 1 & 1 & 0 \\ 3 & -1 & 1 & 0 \\ 0 & 0 & 2 & -1 \\ 0 & 0 & 1 & 0 \\ 0 & 0 & -1 & 3 \end{bmatrix}$$

For a Jacobian $\boldsymbol{J}$ displayed in a basis aligned with the product decomposition $\mathfrak{B} = \bigoplus_i T_{\boldsymbol{s}_i} \mathcal{S}_i$, let $\|\boldsymbol{J}\|_0$ denote the number of its nonzero entries. In this aligned basis, we have

$$\rho_{\mathfrak{B}}^+ \leq \|\boldsymbol{J}\|_0.$$

For a (right) change of source basis $\boldsymbol{G} \in \mathrm{GL}(T_{\boldsymbol{s}}\mathcal{S})$ that *respects* $\mathfrak{B}$, the transformed Jacobian is $\boldsymbol{JG}$, and

$$\rho_{\mathfrak{B}}^+ = \min_{\boldsymbol{G} \in \{\textit{block-diagonal}\}} \|\boldsymbol{JG}\|_0.$$

Conversely, for a change of basis $\boldsymbol{G} \in \mathrm{GL}(T_{\boldsymbol{s}}\mathcal{S})$ that does not *respect* $\mathfrak{B}$,

$$\rho_{\mathfrak{B}}^- = \inf_{\boldsymbol{G} \notin \{\textit{block-respecting}\}} \|\boldsymbol{JG}\|_0.$$

Likewise, for a single component $i$ with a (nontrivial) split $\mathfrak{B}_i = U \oplus V = T_{\boldsymbol{s}_i} \mathcal{S}_i$, we compare $\rho_{\mathfrak{B}_i}^+$ vs. $\rho_{\mathfrak{B}_i}^-$ using changes of basis that do (or do not) respect $\mathfrak{B}_i$ while fixing basis elements spanning $\bigoplus_{j \in [K] \setminus \{i\}} T_{\boldsymbol{s}_j} \mathcal{S}_j$.

**Mechanistic independence and irreducibility of Type M.** *Since no column support contains or is contained in the support of a column from a different block, Type M mechanistic independence holds in all cases. For $\boldsymbol{A}, \boldsymbol{B}, \boldsymbol{C}$, each component is one-dimensional, so Type M irreducibility holds vacuously. The first block of $\boldsymbol{D}$ is further reducible since $\boldsymbol{D}_{:,1} \pitchfork \boldsymbol{D}_{:,2}$ while the second block is irreducible as $\boldsymbol{D}_{:,3} \supset \boldsymbol{D}_{:,4}$. Note that in the sparsest product-splitting basis multi-dimensional factors cannot be Type M reducible.*

**Irreducibility of Type S.** *Again, since each component of $\boldsymbol{A}, \boldsymbol{B}, \boldsymbol{C}$ is one-dimensional, Type S irreducibility holds automatically. For the Jacobian $\boldsymbol{D}$, each component is two-dimensional; thus we must verify that no 2D block can be internally split to reduce sparsity compared with all other possible splits.*

First block (columns 1–2). *Consider the displayed split $\mathfrak{B}_1 = T_{\boldsymbol{s}_1} \mathcal{S}_1$ and any other nontrivial internal split $\widetilde{\mathfrak{B}}_1 = U \oplus V$. Since both $U$ and $V$ are one-dimensional, no further $\mathfrak{B}_1$-respecting basis transformation can reduce the support. Counting nonzeros yields $\rho_{\mathfrak{B}_1}^+ = 8$, and since*

$$\boldsymbol{G} = \begin{bmatrix} 1 & 0 \\ 1 & 1 \end{bmatrix}, \quad \|\boldsymbol{D}_{:,\{1,2\}}\boldsymbol{G}\|_0 = \|\boldsymbol{D}_{:,\{1,2\}}\|_0,$$

*we obtain $\rho_{\mathfrak{B}_1}^+ = \rho_{\mathfrak{B}_1}^-$.*

*For a distinct split $\widetilde{\mathfrak{B}} \neq \mathfrak{B}$, we have $\rho_{\widetilde{\mathfrak{B}}_1}^- \leq \rho_{\mathfrak{B}_1}^+$, since we can always revert to the current split. Moreover, $\rho_{\widetilde{\mathfrak{B}}_1}^+ \geq \rho_{\mathfrak{B}_1}^+$, as the current split already achieves minimal support. Hence, the first block is irreducible. (We could construct an alternative Jacobian with reducible first component by setting both $-1$ entries in $\boldsymbol{D}$ to $0$; modifying only one is insufficient.)*

Second block (columns 3–4). *Here a local simplification is possible: by mixing the third and fourth columns appropriately, we can reduce the third column by one nonzero. After this adjustment, the same argument as above shows that the second block is also Type S irreducible.*

**Mechanistic independence of Type S.** *We now check mechanistic independence for each Jacobian individually.*

Case $\boldsymbol{A}$. *Columns (blocks) have* exclusive rows*: rows $1, 2$ are nonzero only in the first block, and rows $7, 8$ only in the second. Any non-respecting change of basis mixes the two one-dimensional components, introducing nonzeros into these exclusive rows while at most one of the four shared rows in the middle can be canceled. Thus, any genuine mixing strictly increases the total $\ell_0$-norm, so $\rho_{\mathfrak{B}}^- > \rho_{\mathfrak{B}}^+ = \|\boldsymbol{A}\|_0$.*

Case $\boldsymbol{B}$. *Pairwise, $\boldsymbol{B}$ behaves analogously to $\boldsymbol{C}$: for each column pair there are four exclusive rows and only one shared. This enforces a lower bound under any $2 \times 2$ mix, so all pairwise checks pass.*

*However, there exists a full $\boldsymbol{G} \in \mathbb{R}^{3\times3}$ mixing all three columns without increasing the overall support (thus violating strict inequality in Def. 17):*

$$\boldsymbol{G} = \begin{bmatrix} 1 & 0 & 1 \\ 1 & 1 & 0 \\ 1 & 0 & 0 \end{bmatrix}, \quad \|\boldsymbol{BG}\|_0 = \|\boldsymbol{B}\|_0.$$

*Hence, $\boldsymbol{B}$ is pairwise but not fully mechanistically independent.*

Case $\boldsymbol{C}$. *As in $\boldsymbol{B}$, all pairwise checks pass. The key difference is that in $\boldsymbol{C}$ the three shared rows (1st, 3rd, 5th) cannot be simultaneously eliminated by any invertible $\boldsymbol{G} \in \mathbb{R}^{3\times3}$. Thus, any combination involving all three blocks necessarily preserves the three exclusive rows (2nd, 4th, 6th) and increases $\ell_0$. Therefore, $\rho_{\mathfrak{B}}^- > \rho_{\mathfrak{B}}^+ = \|\boldsymbol{C}\|_0$, i.e., $\boldsymbol{C}$ is fully mechanistically independent.*

Case $\boldsymbol{D}$. *A local simplification inside the second block (mixing the third and fourth columns) reduces the third column by one nonzero. After this, the first, second, and third columns each have four nonzeros (the fourth remains at two), giving $\rho_{\mathfrak{B}}^+ = 14 = 4 + 4 + 4 + 2$.*

*To break Type S independence, one would need a cross-block mix: there must exist a vector $(a, b, c, d)$ with either $a$ or $b$ nonzero and either $c$ or $d$ nonzero such that*

$$\boldsymbol{D}\,(a, b, c, d)^\top$$

*has at most four nonzero entries (matching $\rho_{\mathfrak{B}}^+$). This is impossible: any such combination has at least five nonzeros, even under careful cancellations. Hence, every cross-block mixing strictly increases the $\ell_0$-norm, and $\boldsymbol{D}$ is Type S mechanistically independent.*

*In summary, all components of $\boldsymbol{A}, \boldsymbol{B}, \boldsymbol{C}, \boldsymbol{D}$ are Type S irreducible; $\boldsymbol{A}, \boldsymbol{C}, \boldsymbol{D}$ are Type S mechanistically independent; $\boldsymbol{B}$ is pairwise but not fully mechanistically independent.*

**Example 2** (Minimizers of compositional contrast yield Type S independence in some generators). *This example shows that there exist generators whose latent components are Type S independent but not Type D independent, yet for which the compositional contrast $C_{\text{comp}}$ recovers the sources up to permutation and element-wise transformations.*

*Let $\boldsymbol{s} \in \mathbb{R}^2$ and $\boldsymbol{g} \colon \mathbb{R}^2 \to \mathbb{R}^5$ with $\boldsymbol{g}(\boldsymbol{s}) = \boldsymbol{As}$, where*

$$\boldsymbol{A} = \begin{pmatrix} 1 & 0 \\ 1 & 0 \\ 1 & 1 \\ 0 & 1 \\ 0 & 1 \end{pmatrix}.$$

*We immediately observe that $s_1$ and $s_2$ are Type S but not Type D independent.*

*Consider now a learned decoder $\hat{\boldsymbol{g}} \colon \mathbb{R}^2 \to \mathbb{R}^5$. If $\hat{\boldsymbol{g}}$ minimizes the reconstruction error, then its Jacobian at some $\boldsymbol{z}^* \in \mathbb{R}^2$ takes the form $\boldsymbol{J}_{\hat{\boldsymbol{g}}}(\boldsymbol{z}^*) = \boldsymbol{AB}$ for a nonsingular matrix $\boldsymbol{B}$. Equivalently, $|\det(\boldsymbol{B})| \geq q$ for some $q > 0$. Writing*

$$\boldsymbol{B} = \begin{pmatrix} a & b \\ c & d \end{pmatrix},$$

*we obtain*

$$\boldsymbol{AB} = \begin{pmatrix} a & b \\ a & b \\ a+c & b+d \\ c & d \\ c & d \end{pmatrix} \quad \text{and} \quad C_{\text{comp}}(\boldsymbol{B}) = 2|a||b| + 2|c||d| + |a+c||b+d|.$$

*We will show that*

$$\min_{|\det(\boldsymbol{B})| \geq q} C_{\text{comp}}(\boldsymbol{B}) = q,$$

*and that every global minimizer of $C_{\text{comp}}$ is a generalized permutation matrix (i.e., a matrix with exactly one nonzero entry in each row and each column). This means that the learned latent factors are Type S independent after joint minimization of the reconstruction error and $C_{\text{comp}}$.*

*Proof.* We prove the claim in three steps.

**Step 1.** *Reduction to the case* $|\det(\boldsymbol{B})| = q$.

For $t > 0$ we get

$$C_{\text{comp}}(t\boldsymbol{B}) = t^2 C_{\text{comp}}(\boldsymbol{B}), \qquad |\det(t\boldsymbol{B})| = t^2 |\det(\boldsymbol{B})|.$$

If $|\det(\boldsymbol{B})| > q$, choose $t = \sqrt{q/|\det(\boldsymbol{B})|} < 1$. Then

$$|\det(t\boldsymbol{B})| = q, \qquad C_{\text{comp}}(t\boldsymbol{B}) = t^2 C_{\text{comp}}(\boldsymbol{B}) < C_{\text{comp}}(\boldsymbol{B}).$$

Thus any minimizer must satisfy $|\det(\boldsymbol{B})| = q$. It therefore suffices to prove

$$C_{\text{comp}}(\boldsymbol{B}) \geq |\det(\boldsymbol{B})| \qquad \text{for all } \boldsymbol{B},$$

and to identify the matrices for which equality holds.

**Step 2.** *A chain of inequalities.*

Let

$$x = |a|, \qquad y = |b|, \qquad u = |c|, \qquad v = |d|.$$

By the triangle inequality,

$$|a + c| \geq |x - u|, \qquad |b + d| \geq |y - v|.$$

Hence,

$$C_{\text{comp}}(\boldsymbol{B}) \geq 2xy + 2uv + |x - u| \cdot |y - v| \ =: \ F(x, y, u, v).$$

We now claim

$$F(x, y, u, v) \geq xv + yu \quad \text{for all } x, y, u, v \geq 0.$$

Define

$$D := F(x, y, u, v) - (xv + yu) = 2xy + 2uv + |x - u||y - v| - xv - yu.$$

We analyze $D$ by cases on the signs of $x - u$ and $y - v$.

*Case 1: $x \geq u$ and $y \geq v$.* Then $|x - u| = x - u$, $|y - v| = y - v$, and

$$\begin{aligned}
D &= 2xy + 2uv - xv - yu + (x - u)(y - v) \\
&= 3xy + 3uv - 2xv - 2yu \\
&= (xy + uv) + 2(x - u)(y - v) \ \geq \ 0.
\end{aligned}$$

*Case 2: $x \geq u$ and $y < v$.* Then $|x - u| = x - u$, $|y - v| = v - y$, and

$$\begin{aligned}
D &= 2xy + 2uv - xv - yu + (x - u)(v - y) \\
&= 2xy + 2uv - xv - yu + (xv - xy - uv + uy) \\
&= xy + uv \ \geq \ 0.
\end{aligned}$$

*Case 3: $x < u$ and $y \geq v$.* By symmetry with Case 2 (interchanging $(x, u)$), we again obtain

$$D = xy + uv \geq 0.$$

*Case 4: $x < u$ and $y < v$.* Then $|x - u| = u - x$, $|y - v| = v - y$, and

$$\begin{aligned}
D &= 2xy + 2uv - xv - yu + (u - x)(v - y) \\
&= 2xy + 2uv - xv - yu + (uv - uy - xv + xy) \\
&= 3xy + 3uv - 2xv - 2yu \\
&= (xy + uv) + 2(x - u)(y - v) \ \geq \ 0.
\end{aligned}$$

In all cases we have $D \geq 0$, so indeed

$$F(x, y, u, v) \geq xv + yu = |a||d| + |b||c|.$$

Finally, the determinant satisfies

$$| \det(\boldsymbol{B})| = |ad - bc| \leq |ad| + |bc| = |a||d| + |b||c| = xv + yu$$

by the triangle inequality. In summary, we have the chain

$$C_{\mathrm{comp}}(\boldsymbol{B}) \;\geq\; F(x, y, u, v) \;\geq\; xv + yu \;\geq\; |\det(\boldsymbol{B})|.$$

In particular, if $|\det(\boldsymbol{B})| = q$, then

$$C_{\mathrm{comp}}(\boldsymbol{B}) \geq q.$$

**Step 3.** *Equality conditions and classification of minimizers.*

To attain the minimum under $|\det(\boldsymbol{B})| \geq q$, we must have $|\det(\boldsymbol{B})| = q$ and equality throughout the chain

$$C_{\mathrm{comp}}(\boldsymbol{B}) \geq F(x, y, u, v) \geq xv + yu \geq |\det(\boldsymbol{B})|.$$

*(i) Equality in $C_{\mathrm{comp}}(\boldsymbol{B}) \geq F(x, y, u, v)$.*

We used

$$|a + c| \geq \big||a| - |c|\big|, \qquad |b + d| \geq \big||b| - |d|\big|.$$

This requires

$$ac \leq 0, \qquad bd \leq 0.$$

*(ii) Equality in $F(x, y, u, v) \geq xv + yu$.*

From the case analysis above, equality $F(x, y, u, v) = xv + yu$ forces

$$xy = 0 \quad \text{and} \quad uv = 0,$$

that is,

$$|a||b| = 0, \qquad |c||d| = 0,$$

so

$$\text{either } a = 0 \text{ or } b = 0, \quad \text{and} \quad \text{either } c = 0 \text{ or } d = 0.$$

*(iii) Equality in $xv + yu \geq |\det(\boldsymbol{B})|$.*

We used

$$|\det(A)| = |ad - bc| \leq |ad| + |bc| = xv + yu,$$

which requires

$$(ad)(bc) \leq 0.$$

From (ii) we get four structural patterns, two of which are incompatible with $\det(\boldsymbol{B}) \neq 0$. The remaining possibilities are

$$\boldsymbol{B} = \begin{pmatrix} a & 0 \\ 0 & d \end{pmatrix} \quad \text{or} \quad \boldsymbol{B} = \begin{pmatrix} 0 & b \\ c & 0 \end{pmatrix}.$$

For these matrices,

$$C_{\mathrm{comp}}(\boldsymbol{B}) = |\det(\boldsymbol{B})| \in \{\, |ad|, \ |bc| \,\},$$

so equality holds everywhere.

These are precisely the $2 \times 2$ generalized permutation matrices. Consequently, the global minimizers of $C_{\mathrm{comp}}$ under the constraint $|\det(\boldsymbol{B})| \geq q$ are exactly the generalized permutation matrices with $|\det(\boldsymbol{B})| = q$, and the minimum value of $C_{\mathrm{comp}}$ is $q$. $\qquad \square$

## C    DISENTANGLEMENT FOR NON-INVERTIBLE GENERATORS

In some applications the underlying generator is non-invertible when modelling the latent space as a product space (or a subset thereof). For two such scenarios we can nevertheless make meaningful statements about disentanglement: (1) *local invertibility*, and (2) *invertibility on an open subset*.

For the former category, an example is image data containing multiple objects with identical appearance, since the generator is then (block-)permutation invariant. Another example is the angle of a rotary joint with multiple revolutions in a robotic arm, as $\theta + n(2\pi)$ maps to the same physical state. More generally, these situations involve symmetries such as permutation or rotational symmetry.

For the second category, an example arises from occlusions in image data. Here the generator is also non-invertible, but now entire regions of the latent space map to the same observation (e.g., when one object is hidden behind another).

In such cases multiple latent codes map to the same observation, which makes the encoder inherently ambiguous. The learning algorithm must make a choice about how to represent those observations. This choice may lead to defects in the encoder, such as discontinuities (as discussed later). However, a decoder need not suffer from this issue. As long as we can learn a decoder that generates the observation manifold and whose components are mechanistically independent, we can still obtain disentanglement in case (1), and at least on the invertible subsets in case (2).

We now consider two illustrative examples.

**Example 1.**    Consider images depicting two balls of identical appearance at arbitrary positions in the image, but without occlusion. In this case the latent space can be modelled using an ordered configuration space, representing tuples of pairwise distinct object states:

$$\mathrm{Conf}_K(\Omega) \coloneqq \left\{ \boldsymbol{s} \in \Omega^K \mid \boldsymbol{s}_i \neq \boldsymbol{s}_j, \, \forall\, i \neq j \in [K] \right\}, \tag{24}$$

where $\Omega$ denotes the state space of a single object (e.g., position, color), and $K$ is the number of objects. Since any permutation of the factors (i.e., objects) yields the same observation, the ground-truth decoder must be permutation invariant. The observation manifold can therefore be viewed as an *unordered* configuration space, obtained by quotienting out permutations.

Assuming a soft rasterizer, the generator can be modelled as a local diffeomorphism: the map is locally invertible, and small latent perturbations produce small and reversible changes in the observations. A direct check verifies that we have Type D independence and, by implication, also Type M/S/$H_n$ independence. Because a single ball cannot be itself represented as an additive function with two or more components, we have Type $H_2$ irreducibility, and thus also Type D irreducibility. The generator is also second-order separable, since all first- and second-order partial derivatives are linearly independent at every point in the latent space. Considering the affine equivariance of positional encoding, the generator must also be Type S irreducible. Thus the local identifiability results for Type D, S, and $H_2$ apply.

Furthermore, any configuration space with $K \geq 2$ and $\dim(\Omega) \geq 2$ is connected. Its 1-slices are also connected: fixing one ball, the other can be moved continuously to any other position in the image while avoiding collisions. Hence Theorem 1 applies, and local disentanglement extends to global disentanglement.

**Example 2.**    Consider images of two balls, one large and one small, placed at different locations, with possible occlusion (the smaller potentially disappearing behind the larger). The latent space can be modelled as $\mathcal{S} = \mathbb{R}^2 \times \mathbb{R}^2$, describing the $(x, y)$-positions of both balls. The generator then maps a hyper-tube of latent codes (corresponding to positions of the smaller ball behind the larger one) to the same observations. With a soft rasterizer, the generator becomes differentiable, but it is not invertible (not even locally invertible) on the full domain. However, at any point outside the hyper-tube it *is* locally invertible.

Following the same reasoning as in Example 1, we obtain local identifiability of the decoder at all points outside the hyper-tube. Restricted to this region, the model is even globally identifiable, since the resulting space is identical to that of the previous example. Therefore, if we train a decoder with mechanistically independent components that can generate the observation manifold, it must be globally disentangled outside the hyper-tube.

Whether disentanglement holds *within* the hyper-tube is undecidable: if the large ball occludes the smaller one and moves by a small amount, we cannot determine from the observation alone whether the smaller ball behind it moved as well.

## D    Experimental Details and Further Experiments

### D.1    Compositional Contrast as a Surrogate for Type S Independence

This experiment closely follows the setup of Brady et al. (2023). We first sample latent variables from a standard normal distribution and then generate observations by passing them through an invertible MLP. The outputs are concatenated as

$$\boldsymbol{g}(\boldsymbol{s}) = \big(\boldsymbol{g}^{(1)}(\boldsymbol{s}_1), \boldsymbol{g}^{(1,2)}(\boldsymbol{s}_1, \boldsymbol{s}_2), \boldsymbol{g}^{(2)}(\boldsymbol{s}_2), \boldsymbol{g}^{(2,3)}(\boldsymbol{s}_2, \boldsymbol{s}_3), \dots, \boldsymbol{g}^{(K)}(\boldsymbol{s}_K)\big).$$

For each $\boldsymbol{g}^{(i)}$, the slot dimension is fixed at $\dim(\mathcal{S}_i) = 3$, and the slot-output dimension is set to 20. The overlap ratio is determined by the output dimensions of $\boldsymbol{g}^{(1)}$ and $\boldsymbol{g}^{(1,2)}$: if they have the same number of output dimensions, the overlap is 50%. Strictly speaking, for $K > 2$, this implies that in Figure 2, $K - 2$ slots exhibit a 66% overlap.

We train models with $K \in \{2, 3, 5\}$ slots and regularization parameters $\lambda \in \{10^{-2}, 1\}$, where the loss is $\mathcal{L} = \mathcal{L}_{\text{recon}} + \lambda C_{\text{comp}}$. For each configuration, we run five random seeds across overlap levels $\{0\%, 5\%, 20\%, 50\%\}$, resulting in 120 models in total. To ensure comparability across different numbers of slots and regularization parameters, we apply the same normalization procedure to all experiments. In addition, within each group of models sharing the same overlap ratio, we normalize $C_{\text{comp}}$ by dividing by the group mean, since the achievable minimum of $C_{\text{comp}}$ varies substantially with overlap.

### D.2    Experiments on Non-Invertible Generators

We now consider images depicting two balls whose colors lie between green and red and which may appear at any position in the image, but without occlusion ($K = 2$, $d_s = 6$); see Figure 4. We train an autoencoder with an *additive decoder* (i.e., Type $H_2$ independence), defined as

$$\hat{\boldsymbol{g}}(\boldsymbol{z}) = \sum_{i \in [K]} \hat{\boldsymbol{g}}^{(i)}(\boldsymbol{z}_i),$$

using the standard Mean Squared Error (MSE) reconstruction loss (implementation details are provided below).

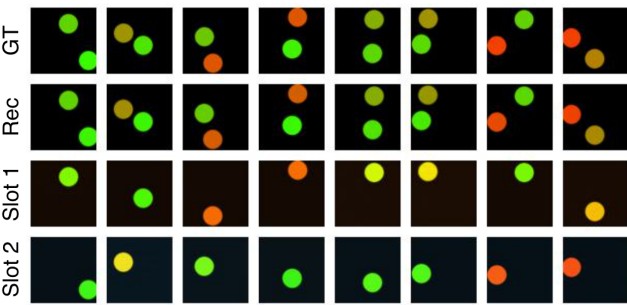

Figure 4: Image reconstructions for the entire latent code and for individual slots. Note that the per-slot reconstructions appear brighter because the offset is undetermined in an additive model.

To evaluate disentanglement quantitatively in this setting, we cannot directly predict the ground-truth latent code from the learned representation using the Slot Identifiability Score (SIS) as before. Because multiple latent codes always correspond to the same observation, the prediction target is ambiguous. Instead, we first determine the best-fitting *fundamental domain* of the latent space under permutations. A fundamental domain is any connected subset of the configuration space containing

exactly one representative of each latent orbit under permutations. Restricting the generator to a fundamental domain renders it invertible, making the prediction target unique.

However, there exist infinitely many choices of fundamental domains, and for most of them the learned regressor would need to approximate a discontinuous function. To avoid this, for each reconstructed image we compute the centers of mass of both balls in image space and select, among all permutations of the ground-truth factors, the closest match. This produces a partition of the latent space that aligns as closely as possible with the encoder's (arbitrary) convention. We then compute the SIS on this selected fundamental domain and denote the resulting metric by SIS*. Table 1 shows that the model achieves nearly perfect disentanglement.

Table 1: Slot Identifiability Scores after selecting the best-fitting fundamental domain over 5 random seeds.

| RMSE | SIS* |
|---|---|
| $1.30 \pm 0.18$ | $99.6 \pm 0.05$ |

Next, we examine the encoder in more detail. As noted by Zhang et al. (2019); Hayes et al. (2023), the encoder must approximate discontinuities in this setting, a phenomenon known as the *responsibility problem*. Such discontinuities necessarily arise whenever we traverse a path in the latent space that connects a point to its block-permutation.

Figure 5 shows the learned latent variables for several latent traversals: in each row, only one ground-truth latent variable is varied while the others remain fixed. The corresponding images and reconstructions are shown in Figure 6. When we vary the coordinates of the first ball $s_1$, only the second encoded slot $z_2 = (z_{2,1}, z_{2,2}, z_{2,3})^\top$ changes; modifying $s_2$ analogously affects only $z_1$, with one exception: The lower-left traversal in Figure 5 reveals an (approximately) discontinuous jump in the middle. On either side of this jump, changes to the ground-truth latent affect only one slot. This discontinuity is unavoidable and can, in principle, hinder autoencoder training by trapping the optimization in poor local minima. However, in this instance, training succeeds without issue.

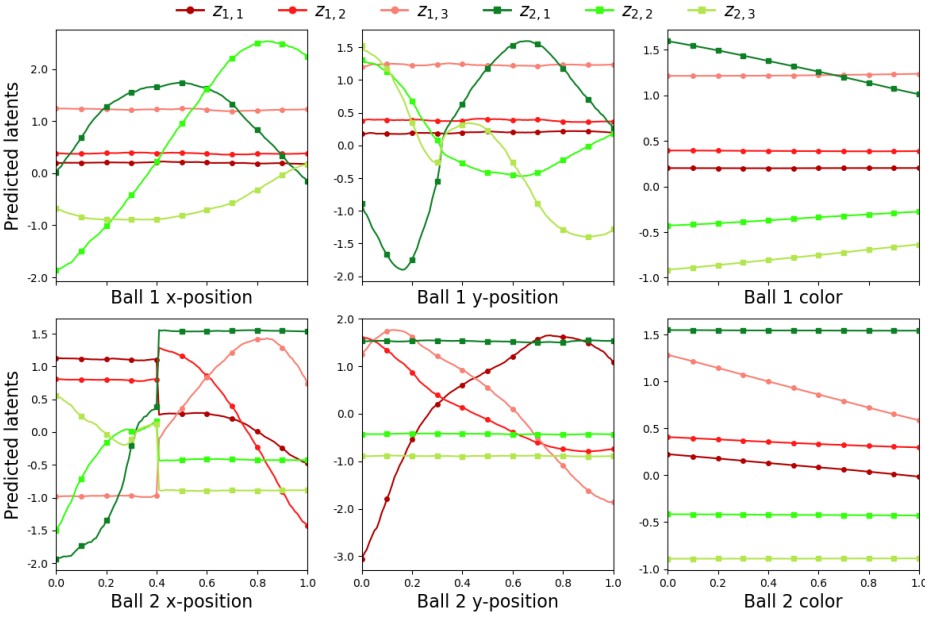

Figure 5: Latent traversals: in each subplot, one ground-truth latent variable varies while all others remain fixed. The curves depict the learned latent codes.

In Figure 6, consider the traversal of the $x$-position of the second ball: the image reconstructions align perfectly with the ground truth, and nothing in the visual output betrays the latent discontinuity.

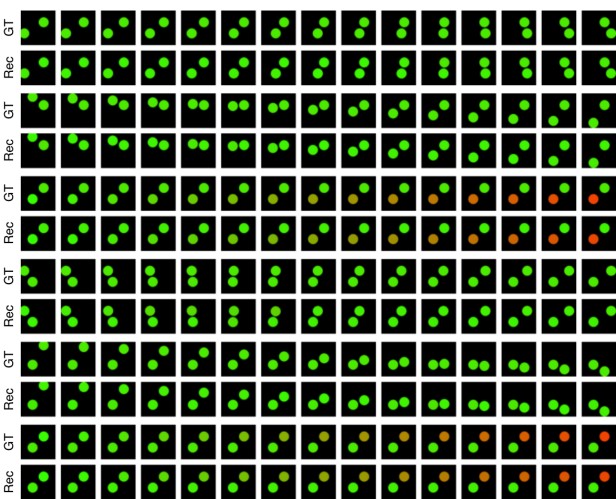

Figure 6: Image reconstructions for latent traversals: in each row, a single ground-truth latent variable is varied while all others remain fixed. From top to bottom, we vary the $x$-position, $y$-position, and color of the first ball, followed by the same for the second.

## Implementation Details

**Dataset.** We use images of size $64 \times 64 \times 3$ with pixel values in $[0, 255]$. The dataset contains 300,000 training images, 10,000 validation images, and 50,000 test images. Images with occlusions are removed, introducing mild dependencies between object positions. Apart from this, the latent variables are sampled uniformly over their support.

**Model Architecture.** The *encoder* consists of:

(1) A ResNet-18 backbone with the final classification layer removed (output dimension: 512).

(2) A linear layer mapping $512 \rightarrow 4096$, followed by Batch Normalization and a Leaky ReLU (slope $0.01$ for negative inputs).

(3) A fully connected layer of size $4096 \times 4096$, again followed by Batch Normalization and a Leaky ReLU.

(4) A final linear layer mapping 4096 to the total ground-truth latent dimension, followed by Batch Normalization.

We use an *additive decoder* $\hat{g}(z) = \sum_{i \in [K]} \hat{g}^{(i)}(z_i)$, where each subdecoder $\hat{g}^{(i)}$ has the same architecture (with no shared weights):

(1) A linear layer mapping from $d_i = \frac{d_s}{K}$ to 1024, followed by Batch Normalization and a Leaky ReLU.

(2) Four fully connected layers of size $1024 \times 1024$, each followed by Batch Normalization and a Leaky ReLU. The output is reshaped into 64 feature channels over a $4 \times 4$ grid.

(3) A stack of deconvolutional layers, each followed by a Leaky ReLU:

(a) Deconvolution: $64 \rightarrow 1024$, kernel size 4, stride 2, padding 1.

(b) Deconvolution: $1024 \rightarrow 512$, kernel size 4, stride 2, padding 1.

(c) Deconvolution: $512 \rightarrow 128$, kernel size 4, stride 2, padding 1.

(d) Deconvolution: $128 \rightarrow 3$, kernel size 4, stride 2, padding 1.

**Hyperparameters.** We use the AdamW optimizer (Loshchilov & Hutter, 2017) with:

- Batch size: 1024,
- Learning rate: $5 \times 10^{-5}$,
- Weight decay: $1 \times 10^{-5}$,
- Number of training epochs: 1000.

## LLM USAGE DISCLOSURE

In accordance with the ICLR policy on large language model (LLM) usage, we disclose that an LLM (OpenAI's ChatGPT) was used solely for minor language polishing. This included limited grammar correction and rephrasing for clarity. All research ideas, technical content, analyses, and conclusions were generated entirely by the authors, who remain fully responsible for the paper's content. For full transparency, this very disclosure note was also drafted with the help of ChatGPT.

