# OpenReview forum: "Mechanistic Independence: A Principle for Identifiable Disentangled Representations"
_ICLR.cc/2026/Conference — ICLR 2026 Poster_

### Official Review · Reviewer_Crp8 · 2025-10-31

**Soundness:** 3
**Presentation:** 3
**Contribution:** 3
**Rating:** 6
**Confidence:** 3

**Summary:**

This paper proposes mechanistic independence, which enables identifiable disentangled representations without strong statistical assumptions.

**Strengths:**

1. As a theoretical paper, the presentation is rigorous.
2. The theorem proofs appear solid.
3. The authors seem familar with recent identifiability literature and compare their results with prior work across the types D, M, S, H.

**Weaknesses:**

1. The abstract and main text repeatedly state that the paper’s identifiability results allow a non-invertible g, and they give the practical example of the “responsibility problem” (lines 160–161) to justify relaxing the global invertibility assumption. From my reading, permitting non-invertibility appears to be one of the paper’s contributions/novelties. However, all of the identifiability results actually rely on local diffeomorphism, which is stronger than local invertibility. Therefore, I think it is necessary to justify that this relaxation is non-trivial. By “trivial,” I mean the situation where one advertises that the framework does not require Condition A, but actually assumes Condition A holds in all the relevant region in the space (i.e. only not hold on a set of measure 0). It would be nice if you can add a discussion clarifying the gap between global invertibility and local diffeomorphism. For example, could you analyze a practical case like the “responsibility problem” that fails to be globally invertible on the space of interest but still satisfies the local-diffeomorphism condition?

2. In the theoretical framwork of this paper, S is the souce factors space, Z is the representation/target factor space, X is the observational space. But before def 1 (line 120-123), you only introduce S and X. Notation Z and term 'target factor' first appear in def 1 without any explanation. Maybe you can introduce Z and target factor before def 1 like what you have done with S and X. I think this may make the paper easier to read.



3. The theorem numbering in the appendix does not match the numbering in the main text.

**Questions:**

See Weaknesses.

---

> ### Author Response · Authors · 2025-11-28
>
> We thank the reviewer for the positive assessment of the rigor and literature integration.
>
> 1. Gap between global invertibility and local diffeomorphism
>
> We agree and added discussion plus examples in Appendices C and D.2.
>
> 2. Introducing Z before Definition 1
>
> Thank you for this suggestion. We now introduce Z before Definition 1 for consistency with S and X.
>
> 3. Theorem numbering
>
> We corrected the appendix numbering to match the main text.

---

### Official Review · Reviewer_xBHk · 2025-11-01

**Soundness:** 3
**Presentation:** 3
**Contribution:** 3
**Rating:** 4
**Confidence:** 4

**Summary:**

This paper discusses a new set of principles for the identifiability problem of disentangled representations. Distinguished from existing works that usually assume statistical independence, this paper proposes mechanistic independence instead, to achieve identifiability even when the mixing function is non-invertible. Various related mechanistic independence conditions are proposed, most of which advance some branches of existing literatures.

**Strengths:**

1.	This paper proposes a novel perspective for latent variable identification, i.e., consider disentanglement directly as the goal instead of traditional slot/block-wise equivalence. This is intuitive since global invertibility is not required here.
2.	This paper gives a thorough discussion on the proposed mechanistic independence, providing several options for the identifiability principle. Most options have close relation to related works from different branches of identifiability research, and are unified nicely into one framework.

I am particularly interested in the novel results that global invertibility of mixing function is not required in this paper. Since almost all works on identifiability rely on this assumption, this would be an important advancement if its implication is properly discussed. See my question 1 for discussion.

**Weaknesses:**

1.	Almost no experiment is provided in this paper, which is my major concern for this paper. I understand this is mainly a theoretical paper. However, at least experiments on synthetic data should be provided (Fig. 1 is not enough), to validate each of your main theorems. Without experimental results, it is hard to understand the application scenario of different mechanistic independence principles, and reliability of the theorems are also weakened.
2.	Lack of explanation for the applicability of proposed assumptions in real applications. It is nice to have so much different independence principles as choices, but all principles lead to one common learning method (Fig. 1). I think it is infeasible to select one principle by validating each assumption in practice, and lacking customized methods significantly blurs the boundary among different principles.

I do not insist that identifiability works *must* include experimental results. But for this paper, I think it is helpful and necessary to demonstrate the superiority given that large part of results are upgrades of existing works.

**Questions:**

1.	This work does not rely on global invertibility of mixing function, how much is this condition be relaxed? I find that local diffeomorphism is still need. Together with path-connectivity, can global invertibility be derived? If not, please provide examples, and discuss how possible is it for such cases to happen in real applications.
2.	How to determine L (the number of blocks in Z) in practice? Setting $L=1$ clearly leads to trivial results.
3.	In Line 86, how to understand the symbol $D_{ij}^n$? Should “id” be 0 instead?

---

> ### Author Response · Authors · 2025-11-28
>
> We thank the reviewer for recognizing the novelty of the mechanistic perspective and the unification of several identifiability principles.
>
> 1. Need for more experiments
>
> We improved the experimental section. The main paper now states more clearly how Type D and Type S identifiability are tested. We also included an additional Type H_2 experiment in the appendix.
>
> 2. Applicability of assumptions
>
> We will add a discussion on the applicability and limitations of the proposed independence principles. One of the main motivations for this research was that previous identifiability results on object disentanglement in image data immediately break down for more realistic settings. Type H_2 independence requires that the generator/decoder can be expressed as an additive function. Type D independence means no two slots affect the same pixel. These are only valid for very constrained setups. Also having access to paired data where only one latent slot changes in unrealistic for image data. Kinds of sufficient variability assumptions are also problematic as there are typically coordinates of object states which never change in natural paired data such as in videos. In contrast Type M/S independence allows for more interaction between objects (partial occlusion, shadows, reflections).
>
> 3. Local vs global invertibility
>
> Local diffeomorphism plus path-connectivity does not imply global invertibility. We added examples and discussion in Appendices C and D.2.
>
> 4. How to choose L
>
> Ideally, one would set $L = K$ (and in practice, choosing $L$ slightly larger often works well). Unfortunately, there is no universal rule for selecting $L$ when $K$ is unknown. In practice, we simply try several values; this is the common approach across identifiability research.
>
> How to interpret the condition $L \leq K$ in the theorems:
> It states that if, for example, the true source space has $K = 3$ factors with dimensions $2$, $3$, and $4$, and the learned representation has $L = 2$ factors with dimensions $5$ and $4$, then $z_1$ must encode $s_1$ and $s_2$, while $z_2$ must encode $s_3$. It would be impossible, for example, for $s_3$ to be partially encoded in both $z_1$ and $z_2$.
>
> 5. Use of “id”
>
> Here, “id” denotes the identity map on the corresponding tangent space.
> If $f : \mathbb{R}^2 \to \mathbb{R}^n$, then $T := D^3 f_p$ is a trilinear map $T(u, v, w)$ with $u, v, w \in \mathbb{R}^2$.
> In this notation, $D^{3}_{1,2} f_p(a, b, w) = T\left((a,0), (0,b), w\right)$ with $a,b \in \mathbb{R}, w \in \mathbb{R}^2$.
> If we were to replace the identity map by the zero map, we would instead obtain $T\left((a,0), (0,b), 0\right) = 0 \quad \text{for all } a,b$, which is not the intended meaning.

---

### Official Review · Reviewer_UZdZ · 2025-11-03

**Soundness:** 3
**Presentation:** 3
**Contribution:** 3
**Rating:** 6
**Confidence:** 3

**Summary:**

This paper introduces a unified framework for achieving identifiable disentangled representations through "mechanistic independence", a principle that characterizes latent factors by how they act on observed variables (via the generator) rather than by their statistical distribution. The authors propose several related independence criteria (Type D, M, S, and H_n) and prove that each yields identifiability of latent subspaces under nonlinear, non-invertible mixing. They establish a hierarchy among these criteria and provide a graph-theoretic characterization of latent factors.

**Strengths:**

The novel theoretical perspective shifting from statistical to mechanistic independence is interesting and practically motivated (e.g. by compositionality and transfer learning). This paper is an interesting step connecting disentanglement with discovering the mechanistic structure of data-generating processes.

**Weaknesses:**

The main weakness of the paper is that the theoretical ideas are not especially intuitive. For example, the paper could better discuss when the various independence assumptions are likely to hold in practice. It might help to include more empirical results, not so much because experiments are required to validate the theory, but because experiments might give intuition about settings where the theory is actually relevant and which assumptions hold in which experiment settings.

As a more minor note, the main text would benefit from a short description of the experimental setup (without reading Appendix C, I didn't understand what Figure 1 shows at all).

**Questions:**

N/A

---

> ### Author Response · Authors · 2025-11-28
>
> We thank the reviewer for appreciating the motivation and conceptual value of mechanistic independence.
>
> 1. Intuition behind independence assumptions
>
> We agree and will include a discussion of the applicability and limitations of the different independence Types. Especially for image data there is a nice way to provide intuition by showing how small occlusions, shadows, and reflections etc. break some but leave others intact.
>
> 2. More empirical intuition
>
> We expanded the experimental section which now includes a clearer presentation of Type D and Type S identifiability. We also added an experiment on Type H_2 and permutation-invariant generators to the appendix.
>
> 3. Explanation of the experimental setup
>
> We agree and substantially improved the main-text explanation of the experimental setup and Figure 1.

---

### Official Review · Reviewer_7EYq · 2025-11-03

**Soundness:** 3
**Presentation:** 3
**Contribution:** 2
**Rating:** 4
**Confidence:** 4

**Summary:**

This paper proposes a framework for identifiability of disentangled representation learning based on mechanistic independence. The authors start from properties of the generator from latent variables to observations, rather than from the distribution of latent variables. A family of independence criteria is discussed: Type D (disjoint support via Hadamard orthogonality of Jacobian columns), Type M (mutual non-inclusion of supports in a fixed basis), Type S (a strict sparsity-gap condition on Jacobian representations), and Type H (vanishing cross *n*-th-order derivatives plus *n*-th-order separability). The paper proves identifiability from the local to the global level and provides a graph-theoretic view where factors correspond to connected components. But the assumptions are not tested in experiments or used to guide new methods. Overall, this paper offers a different view of identifiability of disentangled representation learning.

**Strengths:**

- The proof of identifiability is detailed and step-by-step, going further.The proof looks right and kind of makes sense.
- The whole paper is connected by mechanistic independence, which is an interesting view for identifiability.
- Graph-based characterization of factors as connected components aids intuition and potential diagnostics.
- The paper considers a more general generating process than previous works.

**Weaknesses:**

- Several key assumptions are strong, hard to verify, or basis-dependent: Type M depends on a fixed canonical basis; Type S requires the existence and uniqueness of a sparsest product-splitting basis.
- The assumptions are not tested in well-controlled experiments or real-world cases, and the theory doesn't contribute to methods. There is little evidence on real datasets or complex architectures.
- The limitations of mechanistic independence are not specifically discussed.

**Questions:**

- How can these independence criteria be deleted or used in cases such as a synthetic dataset?
- Can mechanistic independence be linked with current theories? Like in intervations: the second derivative for different components $i, j$ is zero like Type M somehow.

---

> ### Author Response · Authors · 2025-11-28
>
> We thank the reviewer for recognizing the clarity of the proofs and the unifying role of mechanistic independence.
>
> 1. Strength of assumptions
>
> We agree partially with the concern. Though the sparsest product-splitting basis need not be unique, and a basis that minimizes Jacobian sparsity always exists. Type S independence states that any cross-slot mixing essentially increases Jacobian density, which we view as a comparatively weak assumption.
> We also note that the duality between independence and reducibility creates an inherent trade-off, suggesting that our framework operates near the limit of what is achievable using purely mechanistic constraints, a perspective we believe to be novel.
>
> 2. Lack of experiments
>
> We have moved the experiment to its own section and clarified that its purpose is specifically to test Type D and Type S identifiability. We also added an experiment on Type H_2 in the appendix. We acknowledge that these experiments rely on synthetic datasets, but we hope the additions help to address the reviewer’s concerns.
>
> 3. Limitations of mechanistic independence
>
> We agree and will add a dedicated limitation section.
>
> 4. When assumptions hold / how they fail
>
> We appreciate this request for intuition. Examples for image data:
> - Type H_2 independence fails if the generator cannot be written additively (i.e. $g(s)\neq\sum_i g^{(i)}(s_i)$). That means no partial occlusions and generally no shadows or reflections are allowed.
> - Type D independence fails when two factors affect the same pixel (e.g. shadows, reflections, transparency, or partial occlusion).
> - Type M independence fails when one factor’s affected pixel set is contained within another’s (e.g., one object is visible only via reflection).
> - Type S independence is more subtle. For one-dimensional slots, it can fail only if the fraction of shared affected pixels across slots exceeds one half (conservative lower bound). For multi-dimensional slots, providing stronger intuition is more difficult.
>
> 5.	Relation to other theories
>
> There may indeed be theoretical connections, although none are immediately evident. We have attempted to point out all relevant links to the best of our ability. We note that the cross-derivative proof technique used in [1] (and in many follow-up works, particularly in causality) and in [2] (as used in this paper for Type H -- assuming the reviewer meant Type H rather than Type M) applies to two different scenarios: paired versus unpaired data. In the paired setting, the technique is applied to the mechanism relating observations, such as an exponential-family conditional distribution. In the unpaired setting, it is applied to the generator or decoder. It is certainly intriguing that the same fundamental idea appears across such different contexts, which may hint at deeper connections. Exploring potential combinations would indeed be interesting future work.
>
> References
>
> [1] Nonlinear ICA of Temporally Dependent Stationary Sources
>
> [2] Additive Decoders for Latent Variables Identification and Cartesian-Product Extrapolation

---

### Official Review · Reviewer_aEKR · 2025-11-03

**Soundness:** 4
**Presentation:** 3
**Contribution:** 4
**Rating:** 8
**Confidence:** 4

**Summary:**

The paper addresses the challenge of identifiability in disentangled representation learning, particularly for nonlinear generative models, by introducing a unified framework based on mechanistic independence. This principle defines latent factors not by their statistical distribution but by how they act on the observation manifold through the generator function.
The authors propose a hierarchy of mechanistic independence criteria (and corresponding irreducibility notions) ranging from
Type D (Disjointness), Type M (Mutual Non-inclusion), Type S (Sparsity Gap), to Type H_n (Higher-Order Separability).
For each criterion, the paper provides identifiability theory. Furthermore, this framework successfully generalizes and unifies several existing mechanistic constraints based or sparsity based identifiability results.

**Strengths:**

Clarity: The paper is very well-written.

Relevance: The paper is very relevant to the field and the contribution is significant. The framework represents a fundamental conceptual advance, providing a robust, distribution-agnostic foundation for identifying latent factors.

Generality: The framework is broad, covering multi-dimensional factors, partial disentanglement, and non-invertible generators. The Theorem 1 result, extending local to global disentanglement under mild topological assumptions, looks like a strong tool.

Unification: By establishing identifiability for a range of mechanistic constraints under nonlinear, non-invertible mixing and statistically dependent latent factors, the paper generalizes and expands several very recent, disparate results.

**Weaknesses:**

The experimental setting looks limited (and not clear to me).  Plus identifying robust surrogate losses for the relaxed criteria, specifically Type M (mutual non-inclusion) and Type S (sparsity gap), remains an open problem. However, given the solid theoretical contribution, these weaknesses are not major.

**Questions:**

Practical Losses for Type M/S: Can the authors propose initial or speculative ideas for a robust surrogate loss that is more reliably minimized in practice?

Independence Hierarchy: Could the authors include in the figure what are the precise conditions under which one criterion is strictly weaker or stronger than another?

Experimental setting: It is not very clear to me that which types of independence are considered in the experiments? By following the setup of Brady et al. (2023) are you saying that Type D is considered? Have you considered any preliminary empirical validation for the other types like $H_n$?

---

> ### Author Response · Authors · 2025-11-27
>
> We thank the reviewer for highlighting the paper’s clarity, generality, and conceptual contribution.
>
> 1. Practical losses for Type M/S
>
> A promising direction may be to formulate independence as a prediction task, similar in spirit to MAE or JEPA. Since Type S independence states that any cross-slot mixing increases Jacobian density, one could view a disentangled representation as one where each slot is explainable from the smallest possible set of observation dimensions (e.g. pixels). This suggests exploring objectives that predict (or reconstruct) a slot from only the image regions it affects, possibly using masked prediction with ViTs. This is however speculative and subject to several practical challenges. This kind of formulation would also have the clear advantage of removing the Jacobian from the loss which would improve scalability.
>
> 2. Independence hierarchy (conditions for implications)
>
> All logical implications in Figure 2 indicated by solid arrows hold without additional assumptions. That means Type D independence always implies all other independence types etc.
> For example:
> - Type D prohibits any pixel in an image from being affected by multiple slots,
> - whereas Type S allows limited overlap (see explanation around line 296).
>
> However, none of our theorems is strictly stronger than another, because identifiability always requires independence + reducibility, and there is a trade-off between them.
>
> 3. Experimental setting (which independence types are tested?)
>
> We agree that the previous presentation was insufficient. We have moved the experiment to its own section and expanded the explanation. Briefly:
> - For 0% overlap, the setup reproduces Type D independence.
> - For >0% overlap, Type D is violated but Type S is still satisfied for the ground-truth generator.
> - In many such cases the global minimizer of $C_{\mathrm{comp}}$ is Type S independent, even though Type D is unachievable (see also Appendix B Example 2).
> - We added an additional experiment for Type H_2 with permutation invariant generator in the appendix.

---

### Author Response · Authors · 2025-12-02

We again thank all reviewers for their time and insightful comments. We believe the revisions including expanded experiments, additional intuition and limitations, clarifications of assumptions, and improved exposition substantially strengthen the paper and address the raised concerns.

---

### Meta-Review · Area_Chair_fFvt · 2026-01-07

**Summary:**

Reviewers raised concerns primarily about the strength and applicability of the paper’s assumptions, the clarity of the experimental setup, and the sufficiency of empirical validation (aEKR, 7EYq, xBHk). Several reviewers noted that the assumptions underlying the theoretical results may be strong and difficult to verify in real-world settings, and requested clearer discussion of when these assumptions are likely to hold (7EYq, xBHk). Reviewer Crp8 highlighted a potential gap between global invertibility and local diffeomorphism assumptions.

The authors addressed these concerns by clarifying the experimental setup (aEKR), adding synthetic experiments, and expanding the discussion on applicability and theoretical assumptions (7EYq, xBHk, Crp8). While some uncertainty remains regarding real-world verification of the assumptions, reviewers generally acknowledged the paper’s solid theoretical contributions.

**Reviewer Concerns:**

### Addressed

1. **Lack of experimental validation:**
   The authors added synthetic experiments to strengthen the empirical evaluation.

2. **Assumptions and theoretical discussion:**
   The authors provided detailed explanations of the assumptions and discussed scenarios in which the assumptions hold or fail.

### Outstanding

1. **Limited real-world applicability:**
   The experimental validation remains limited to synthetic data, and the paper does not demonstrate the applicability of the proposed approach in real-world settings.

**Reviewer Scores:**

Reviewer **aEKR** raised concerns about the clarity of the experimental setting, but explicitly noted that these weaknesses are minor given the paper’s solid theoretical contribution. This reviewer is therefore likely to remain positive.

Reviewer **7EYq** expressed concerns that the assumptions are strong, difficult to verify, and may not be testable in real-world scenarios. In response, the authors added synthetic experiments and provided examples illustrating when the assumptions may hold. This may partially address the concern, and the reviewer may either remain negative or raise their score.

Reviewer **xBHk** similarly noted the lack of experimental validation and insufficient discussion of when the assumptions apply. The authors added synthetic experiments and expanded the discussion on applicability. As a result, this reviewer may remain negative or potentially raise their score.

Reviewer **Crp8** raised concerns about the gap between global invertibility and local diffeomorphism assumptions. The authors added supporting experiments in Appendix C and D.2 to substantiate their claims. This response is likely sufficient for the reviewer to remain positive.

---

### Decision · Program_Chairs · 2026-01-26

Accept (Poster)